# New Approaches in Radiotherapy

**DOI:** 10.3390/cancers17121980

**Published:** 2025-06-13

**Authors:** Matthew Webster, Alexander Podgorsak, Fiona Li, Yuwei Zhou, Hyunuk Jung, Jihyung Yoon, Olga Dona Lemus, Dandan Zheng

**Affiliations:** 1Department of Radiation Oncology, University of Rochester, Rochester, NY 14627, USA; alexander_podgorsak@urmc.rochester.edu (A.P.); fenglifiona_li@urmc.rochester.edu (F.L.); yuwei_zhou@urmc.rochester.edu (Y.Z.); hyunuk_jung@urmc.rochester.edu (H.J.); jihyung_yoon@urmc.rochester.edu (J.Y.); dandan_zheng@urmc.rochester.edu (D.Z.); 2Department of Radiation Oncology, University of Miami, Coral Gables, FL 33146, USA; oxd407@med.miami.edu

**Keywords:** adaptive radiotherapy, advanced image guidance, artificial intelligence and data science, boron neutron capture, brachytherapy, flash radiotherapy, proton radiotherapy, heavy ion radiotherapy, radioimmunotherapy, spatially fractionated radiotherapy, stereotactic radiotherapy, theranostics

## Abstract

Radiation therapy is one of the most common and powerful tools used to treat cancer. Over the past century, it has been continuously evolving into highly advanced treatments that can target tumors very precisely while protecting nearby healthy tissues. In this review, we explore the latest developments that are changing how radiation therapy is delivered and becoming more powerful than ever. These include new imaging tools that allow us to more clearly see the patient’s anatomy, computer systems that can adjust treatments in real time, and smart technologies like artificial intelligence that help plan treatments more accurately. We also look at new ways of treating, such as using ultra-fast doses, more precise delivery methods, and using heavy particles like protons. Other methods combine radiation with the body’s immune system or personalized medicines for better results. These innovations are especially important for treating cancers that are difficult to reach, resistant to standard treatments, or found in children and other sensitive populations. As radiation therapy becomes more targeted and adaptive, it opens the door to more personalized care. By summarizing these cutting-edge approaches, our work supports the ongoing effort to make radiation therapy safer, smarter, and more successful for people facing cancer.

## 1. Introduction

After over a century of research and development in radiotherapy (RT), one might assume that progress toward innovative approaches in RT would be slowing. However, this is far from the case. Over the past century, we have transitioned from ortho- and super-voltage treatment units, which were limited to superficial targets, to the current paradigm of megavoltage X-rays, electrons, and other charged particles that enable the treatment of deeper-seated targets while better sparing the skin surface and nearby normal tissue. We have advanced from having no image guidance for internal anatomy to achieving sub-millimeter accuracy in target localization. Similarly, treatments that once relied on a few static, open radiation beams have evolved into complex arc therapy plans with beam aperture modulation. Despite these remarkable advancements, the boundaries of RT continue to be pushed today.

This review article summarizes current developments in RT, including technological innovations such as advanced image guidance, artificial intelligence (AI) and data science, adaptive RT, and 3D printing, biological and targeting strategies such as radioimmunotherapy and theranostics, and treatment delivery and optimization improvements such as FLASH RT, spatially fractionated radiation therapy (SFRT), and particle therapy. While some of these approaches have already become part of standard clinical practice, others are at the cutting edge and are poised to define the next generation of treatment technologies. Figure 1 shows a schematic of the new RT approaches covered in this review.

Although it is not possible to cover every innovative idea and project currently being explored in the field, this review focuses on the topics that are generating the most interest and have the greatest potential for immediate clinical impact. Specifically, this article outlines the recent advancements and future directions in the following areas:BrachytherapyStereotactic radiotherapyAdvanced image guidanceProton and heavy ion radiotherapyAdaptive radiotherapyHyperthermiaTheranosticsArtificial intelligence and data scienceRadioimmunotherapySpatially fractionated radiotherapyFlash radiotherapyBoron neutron capture

In addition, we included a very brief overview of several more topics which have shown significant promise in RT:Intraoperative radiotherapyVolumetric modulated arc therapy for total body and total marrow irradiationGenomic profiling3D printingAlpha-particle therapyPhotodynamic therapyAuger therapyHydrogen therapy

The ranking of these areas is based on their current level of clinical integration, evidence base, and maturity of technological development, moving from well-established therapies to highly experimental approaches. Those at the top of the list are widely adopted with robust clinical data supporting their use, though they continue to evolve with technological refinements. Mid-tier modalities have demonstrated clinical utility in specific contexts but are not yet universally standardized, with ongoing research expanding their applications. Emerging technologies like genomic-guided radiotherapy and 3D-printed solutions show promise but remain in earlier stages of integration. At the most experimental end are interventions that are still in preclinical or early clinical testing, with significant barriers to widespread adoption. It is important to note that these rankings are not rigid—many fields overlap in maturity, and innovation occurs at all levels, from incremental improvements in established techniques to groundbreaking advances in experimental ones. The dynamic nature of radiation oncology means that today’s experimental approaches may become tomorrow’s standards, particularly as technologies like AI and precision medicine reshape the field.

## 2. Brachytherapy

Brachytherapy, a form of internal RT, involves placing radioactive sources directly into or adjacent to the tumor. It has evolved significantly in recent years, offering more precise and effective treatments while minimizing side effects [1]. After an indolent period for the growth of brachytherapy, emerging technologies have sparked renewed interest and increased utilization of brachytherapy [2]. Many of these areas of interest are represented in Figure 2. Numerous clinical trials are currently underway, exploring a wide range of brachytherapy practices [3]. While this section highlights recent advances, more comprehensive reviews have been published on current and future developments in the field [4,5].

### 2.1. Intensity-Modulated Brachytherapy (IMBT)

Most modern brachytherapy treatments rely on isotropic dose distributions. However, a major area of research focuses on shielded and directional brachytherapy approaches, which achieve superior target coverage while sparing healthy tissues. The trade-offs for these approaches include longer treatment times and increased complexity in delivery.

Most applicators under investigation are designed for high-dose-rate brachytherapy, which typically uses an iridium-192 source. Other applicators utilize lower-energy isotopes or electronic sources to enhance shielding effectiveness. Various intensity-modulated brachytherapy approaches have been proposed for brachytherapy treatments of rectal [6,7], cervical [8,9,10,11], vaginal [12], breast [13], and prostate [14,15] cancers. The challenges associated with IMBT are site-specific, leading to a wide variety of applicator designs. Many designs incorporate static tungsten shields with either multiple catheter channels or a single spiral channel around the shield. More advanced approaches involve dynamic shields capable of rotation and translation, offering greater flexibility in dose delivery.

Several hurdles must be overcome to make shielded brachytherapy approaches viable for clinical use. These include significantly longer treatment times compared to conventional HDR, as well as complexities in miniaturizing shielding, manufacturing radiation sources, and engineering the delivery systems. Advances in 3D printing and novel material technologies may help address these challenges by enabling the creation of more advanced, potentially patient-specific applicators that tailor dose distribution to the individual’s anatomy and tumor profile.

### 2.2. Image-Guided Brachytherapy

The use of image guidance to accurately assess soft tissue has become a cornerstone of global brachytherapy recommendations and guidelines [16,17,18,19,20]. A significant advancement in brachytherapy is the shift from planar imaging and point-based dosimetry to volumetric imaging and volume-based treatment planning. The benefits of MRI guidance have been particularly well-demonstrated in gynecological brachytherapy [21,22]. MRI has been shown to improve implant quality, plan conformity, local disease control, and toxicity reduction [23,24]. Current research is exploring the potential of MRI-guided brachytherapy to increase fractional doses and reduce the total number of treatments [25].

For prostate brachytherapy, transrectal ultrasound (TRUS) has become a critical tool [26,27]. The growing use of three-dimensional (3D) TRUS enables real-time verification and delivery of radiation. In accelerated partial breast irradiation, delineating the tumor bed is essential. Computed tomography is used for tumor bed localization in open-cavity surgeries [28], while ultrasound and mammography play crucial roles in closed-cavity procedures [29].

### 2.3. Three-Dimensional Printing

Improved imaging and soft tissue contrast have enhanced the ability to characterize patient-specific geometries. Three-dimensional printing leverages this capability to create individualized treatment applicators, improving dose conformality, treatment flexibility, and patient comfort.

A simple example of 3D printing in brachytherapy is the creation of custom-sized vaginal cylinders for patients where standard sizes leave air gaps [30,31,32,33]. More complex uses of 3D-printed applicators include fully customized applicator shapes and sizes. Furthermore, these applicators can be incorporated into treatment planning, including optimizing the catheter paths through the applicator, resulting in greater dose conformality [33,34,35]. Similar work has been conducted for many other sites including breast, head and neck [36,37,38], prostate [39], skin [40], and more. Furthermore, the use of 3D printing has enabled treatment of sites not traditionally treated with brachytherapy, such as the pancreas [41].

Clinical studies of 3D-printed applicators for gynecological treatments have shown notable improvements in treatment precision, particularly in achieving superior target coverage (D_90_) when compared to conventional, standardized applicators. The integration of 3D-printed templates has also enhanced the accuracy of implant placement, increased consistency with pre-treatment planning, and enabled the use of more advanced implantation strategies, including non-coplanar configurations and oblique insertion angles [42]. In addition, 3D-printed applicators have been used in head and neck treatments to successfully reduce the number of needles required to treat while maintaining or improving clinical outcomes [43].

### 2.4. Treatment Planning

Treatment planning and dose optimization involve three main steps: anatomic contouring, applicator digitization, and dosimetric planning and optimization. Historically, these steps were performed manually. While contouring remains largely manual, recent research has explored the use of deep learning to improve contouring speed and accuracy [44,45,46,47]. Continued advancements in this area are likely to establish deep learning as a standard tool for contouring.

Applicator reconstruction has evolved from manual placement to highly accurate predefined models integrated into imaging datasets [48]. Recent innovations enable automated applicator reconstruction based on imaging segmentation of quantifiable distortions caused by the applicator. Electromagnetic catheter tracking, which uses passive induction to determine the location and orientation of interstitial needles, has been introduced clinically to improve reconstruction accuracy [49,50,51,52].

Dose calculations are primarily based on the AAPM Task Group 43 formalism [53,54]. However, Monte Carlo and Boltzmann transport calculation methods offer greater accuracy, particularly in the presence of tissue inhomogeneities [55]. These techniques are essential for accurate dose calculation of any shielded applicator. These techniques are essential for accurate dose calculations in shielded applicators and IGBT methods.

Finally, there is growing interest in biological optimization for treatment planning. Several biological models have been implemented to optimize treatment plans based on biological effects [56,57,58].

## 3. Stereotactic Radiosurgery and Stereotactic Body Radiotherapy

The development of Stereotactic Radiosurgery (SRS) and Stereotactic Body Radiotherapy represents a significant advancement in the treatment of brain and small tumors, combining the precision of stereotactic techniques with the therapeutic benefits of focused radiation. SRS was first performed in 1951 by Swedish neurosurgeon Lars Leksell, laying the foundation for its evolution. This highly precise, non-invasive, high-dose RT is used to treat various brain conditions while minimizing damage to surrounding tissues. SRS is widely applied in the management of brain metastases, arteriovenous malformations (AVMs), trigeminal neuralgia, acoustic neuromas, meningiomas, pituitary adenomas, and select gliomas [59,60].

For malignant tumors such as brain metastases, SRS effectively controls tumor growth while preserving cognitive function and minimizing side effects. For non-cancerous conditions like AVMs or trigeminal neuralgia, SRS provides favorable treatment outcomes without the risks associated with invasive surgery.

Emerging in the mid-1990s, SBRT extends SRS principles to extracranial tumors.. It employs high-dose hypofractionated treatment (2–5 fractions), advanced immobilization methods, and intrafraction target motion management to achieve sub-centimeter accuracy, particularly in the lungs, liver, prostate, and spine [61,62,63]. Compared to conventional radiotherapy, SRS and SBRT exhibit high efficacy in treating radioresistant, hypoxic tumors by disrupting the tumor microenvironment, triggering cellular signaling pathways, and modulating the immune response [64]. Today, SRS and SBRT serve as non-invasive alternatives to traditional surgery, offering improved patient outcomes and reduced recovery times.

### 3.1. Treatment Modalities

**Gamma Knife:** The Gamma Knife is a specialized SRS system designed for treating brain lesions. It utilizes multiple highly focused gamma-ray beams from cobalt-60 sources [65]. The latest model, Elekta Esprit, integrates frameless treatment, CBCT, and an infrared motion management system for enhanced precision in patient alignment and movement tracking.

**CyberKnife:** The CyberKnife system features a lightweight linear accelerator mounted on a robotic arm, combined with real-time image guidance, enabling sub-millimeter accuracy. CyberKnife is particularly advantageous for SBRT, as it allows continuous tumor tracking and automatic beam adjustments to compensate for respiratory motion, which is crucial for tumors in the lungs, liver, pancreas, and abdomen [66].

**LINAC-Based SRS and SBRT:** Linear accelerator (LINAC)-based SRS and SBRT are supported by advanced onboard imaging techniques, such as CBCT and MRI [67]. This approach utilizes a linear accelerator to deliver multiple non-coplanar beams of radiation converging at a single isocenter within the body (such as Varian HyperArc) [68]. Innovations such as a six-degree-of-freedom couch and orthogonal X-ray imaging (e.g., BrainLAB ExacTrac) further improve single-isocenter SRS (SI-SRS) accuracy.

The ZAP-X platform is a dedicated LINAC-based radiosurgery system designed with a self-shielded, gyroscopic configuration. Unlike conventional C-arm LINACs, the ZAP-X employs lower beam energy (3 MV) to minimize shielding requirements, variable-sized cylindrical collimation for precise targeting, and a shorter source–axis distance (45 cm) to reduce beam penumbra [69]. In addition to cancer-based indications, ZAP-X is increasingly used for functional neurological disorders such as essential tremor [70].

### 3.2. Treatment Planning Techniques

SI-SRS enables the simultaneous treatment of multiple brain metastases using a single isocenter with a limited field size, reducing rotational target positioning errors. Compared to traditional multi-isocenter SRS, SI-SRS shortens treatment times, improves conformity, reduces dose spillage, and minimizes patient discomfort and setup errors. Advanced planning techniques—including dynamic conformal arc (DCA), VMAT, hybrid approaches (VMAT + DCA), and non-coplanar beam arrangements—enhance dose distribution while minimizing radiation exposure to critical structures [71,72,73].

An emerging approach in treatment planning, biological optimization incorporates radiobiological models, such as tumor control probability and normal tissue complication probability, to improve the balance between tumor control and healthy tissue preservation [74].

Along with improving the planning approach, AI plays a significant role in volume contouring and treatment planning. AI-driven algorithms are revolutionizing volume contouring and treatment planning. By automating these processes, AI significantly reduces planning time and enhances plan quality [75].

Adaptive SBRT represents another significant advancement in RT, allowing for real-time adjustments to treatment plans based on changes in tumor size, shape, or position during the treatment course. This adaptability enhances the precision and effectiveness of SBRT, improving outcomes while minimizing side effects [76].

### 3.3. Future Direction

Dynamic Trajectory Radiotherapy (DTRT) is an emerging technique designed to improve precision and accuracy in SRS and SBRT, particularly for tumors in anatomically challenging locations prone to motion. Unlike standard VMAT, which allows continuous movement of multi-leaf collimator leaves and the gantry, DTRT additionally enables dynamic table and collimator rotations while the beam is active. The collimator angle is optimized to improve target conformity and reduce radiation exposure to OARs. By dynamically adapting to the geometrical and dosimetric characteristics of OARs and the target volume, DTRT ensures accurate dose delivery while minimizing damage to surrounding healthy tissues [77].

AI and machine learning are expected to significantly enhance the efficacy and safety of SRS and SBRT. Beyond automating processes, AI can provide predictive analytics, assessing treatment outcomes and the risk of side effects based on treatment plans and patient-specific health data. This capability allows for more personalized treatment strategies and improved clinical workflow efficiency [75].

Recent studies indicate that integrating SRS with other therapeutic modalities, such as immunotherapy or targeted therapy, may enhance treatment efficacy by leveraging synergistic biological effects [78]. Research into the combined effects of RT and other treatment approaches continues to progress.

Both SRS and SBRT are continually evolving, with ongoing research focused on optimizing treatment protocols, integrating modern treatment technologies, improving imaging techniques, and expanding their applications to a broader range of tumors. Future advancements will likely further enhance precision, patient comfort, and personalized treatment approaches, solidifying SRS and SBRT as critical components of modern oncology.

## 4. Advanced Image Guidance

In recent decades, image guidance has become increasingly utilized during the delivery of conventional radiation therapy. This increased utilization has driven the development of more advanced image guidance techniques, providing the radiation oncology treatment team with enhanced information. IGRT has evolved significantly from the days of weekly portal images [79], which were once the sole form of image guidance during treatment. The introduction of kV-based planar and 3D cone-beam computed tomography (CBCT) imaging marked substantial improvements in imaging during radiation therapy.

Today, image guidance provides volumetric structural data with improved soft-tissue contrast, as well as functional and biological insights. Some methods for obtaining image guidance before, during, or after radiation therapy even avoid the use of ionizing radiation altogether. These advancements aim to achieve the primary goal of radiation therapy: destroying tumor cells while minimizing damage to surrounding normal tissues and organs. One way to achieve this goal is by delivering a more conformal radiation dose to a smaller area around the target using techniques such as intensity-modulated radiation therapy (IMRT). However, with more conformal techniques like IMRT, there is an increased risk of geometric miss compared to treatments using one or a few static open fields. Advanced image guidance can enhance confidence that radiation is being delivered accurately to its intended target.

### 4.1. Magnetic Resonance Imaging

Hybrid imaging-treatment systems represent a promising technology that is becoming more prevalent in modern radiation oncology clinics. One example is the combination of a magnetic resonance imaging (MRI) scanner with a linear accelerator (LINAC), known as the MR-LINAC. The MR-LINAC offers several imaging advantages over traditional LINAC systems paired with on-board CBCT imagers. MRI provides superior soft-tissue contrast compared to CBCT and can acquire image data during radiation delivery, enabling real-time beam gating based on 3D internal imaging of the target relative to the surrounding organs at risk (OARs). Currently, two widely adopted MR-LINAC systems are available, with the primary clinical difference being the MRI field strength. The ViewRay (ViewRay Inc., Cleveland, OH, USA) MRIdian system features a 0.35 T split-bore MRI integrated with a 6 MV LINAC without a flattening filter [80]. In contrast, the Elekta (Elekta AB, Stockholm, Sweden) Unity system combines a 1.5 T MRI scanner with a 7 MV LINAC without a flattening filter [81]. The higher field strength of the Elekta Unity offers potential advantages in image quality and functional MRI capabilities for assessing tumor response during treatment [82]. However, this higher field strength also increases the electron return effect [83]. The ability of the MR-based treatment system to acquire both structural and functional information is an advantage over the PET-based treatment systems that require CBCT for structural information acquisition. Despite these differences, both MR-LINAC systems support ART treatments with the patient on the treatment table. While MR-LINACs can treat a variety of tumor sites, their benefits are particularly pronounced for moving targets in the thorax and abdomen, such as the pancreas, liver, kidneys, and adrenal metastases. Dose escalation for pancreatic targets, guided by real-time MR imaging, has been shown to be safe and associated with favorable long-term outcomes, as demonstrated in a 2024 study [84]. Clinical trials are also exploring the benefits of MR-LINACs for cancers of the esophagus, bladder, brain, lung, rectum, head and neck, prostate, breast, and oligometastatic disease [85,86,87,88,89,90,91,92]. As radiation oncology continues to embrace hypofractionation and margin reduction, the improved image guidance provided by MR-LINACs is expected to become increasingly clinically useful.

### 4.2. Positron Emission Tomography

Positron emission tomography (PET) provides functional information based on the uptake of radioactive tracers, typically a glucose analog bound to fluorodeoxyglucose (FDG), by tumors. PET has become increasingly integrated into the radiation oncology workflow, primarily for pre-treatment staging and post-treatment response assessment. There is substantial clinical evidence supporting the use of FDG-PET for target delineation in head and neck cancers and lung cancers [93,94]. Additionally, other radiotracers have been developed and studied for specific treatment sites. For example, gallium-68 or fluorine-18 bound to prostate-specific membrane antigen (PSMA) has proven valuable for detecting recurrent metastatic prostate cancer. Integrating PET guidance—or biologic guidance—into treatment could transform tumors into fiducials for target localization and tracking, provided PET imaging can be integrated with a treatment unit. The first widely available PET-LINAC system is the RefleXion (RefleXion Medical Inc., Hayward, CA, USA) X1, a biology-guided radiotherapy system that combines a 6 MV flattening filter-free LINAC with a 3D multi-pixel PET scintillation detector. This system has been shown to perform at least as well as a diagnostic-quality PET scanner [95,96]. Studies have demonstrated that the RefleXion system can generate clinically acceptable treatment plans for various sites, including lung, head and neck, anus, prostate, and brain [97]. The RefleXion workflow includes a PET pre-scan before treatment, which generates a predicted dose distribution based on the tumor’s PET signal and target motion. This predicted distribution is compared to the planning-dose distribution to ensure adequate dose delivery to the target and the sparing of critical structures. If the criteria are met, treatment proceeds. Similarly to the MR-LINAC, the PET-LINAC has the potential to improve RT by better compensating for inter- and intra-fraction target motion, enabling tighter margins and eventual dose escalation. Through the use of the PET system, the functional information of the tumor environment is more easily obtained during treatment compared with the MR-based LINAC systems; this functional information can be used for ART to adjust the plan as the target responds to the treatment course. As patients with cancer live longer, the aggressive targeting of oligometastatic disease is becoming increasingly important for advancing radiation oncology [98], and these advanced combined systems for imaging and treatment like the MR-LINAC and PET-LINAC are allowing for further improvement of radiation oncology.

### 4.3. Surface Guidance

Surface-guided radiotherapy (SGRT) refers to the use of optical imaging to acquire and register the real-time 3D surface of a patient relative to a reference surface aligned with the treatment isocenter. SGRT offers unique advantages over other forms of IGRT, particularly for treatment sites where the patient’s surface serves as a reliable surrogate for the target. SGRT has been used for a number of sites, such as cranial targets for SRS, thorax, pelvis, and the breast following lumpectomy [99]. Initially developed to replace tattoos and lasers for patient positioning [100], SGRT’s role has expanded significantly. It is now used for patient identification [101], beam gating during breath-hold treatments of the left breast, and motion assessment during frameless stereotactic radiosurgery [102]. SGRT systems are integrated into all facets of RT delivery. While different vendor solutions vary slightly, they generally rely on the principle of stereovision, where images are acquired simultaneously by at least two cameras from a known geometry. Triangulation of each pixel in the 2D images reconstructs points in 3D space, enabling real-time surface tracking at a rate of >1 frame per second [103]. This is achieved by defining a tracking region of interest, which limits image processing to the surface area corresponding to the target surrogate. Recent studies have extended the use of SGRT beyond breast and brain treatments. For example, breath-hold treatments in the thorax (e.g., lung, lymphoma, mediastinal) can be effectively delivered with surface guidance [104]. Additionally, SGRT has been used for extremity treatments, such as sarcoma or bone metastases, where pain may limit the use of traditional immobilization techniques. In these cases, SGRT has reduced the need for repeated X-ray imaging during treatment [105]. Some SGRT systems also incorporate thermal cameras to augment the available information. Unlike X-ray-based image guidance, SGRT systems do not use ionizing radiation and can therefore be used continuously during treatment for real-time tracking or beam gating. Emerging clinical applications of SGRT include its use during stereotactic body radiotherapy (SBRT) for lung and liver [102], proton-based radiotherapy [106], and even as part of a physicist’s toolkit for machine quality assurance [107].

### 4.4. Cherenkov Radiation

Cherenkov radiation is emitted when charged particles travel faster than the speed of light in a dielectric medium [108]. This phenomenon occurs during RT and can be captured by specialized detectors, enabling real-time visualization of radiation delivery [109,110]. Cherenkov imaging is a powerful tool because the emitted light reflects the shape and location of the treatment beam, with its magnitude proportional to the delivered dose [111,112,113]. It is estimated that approximately 40% of reported radiation delivery incidents are attributed to incorrect patient positioning [114]. In clinical workflows, Cherenkov imaging is acquired using cameras synchronized with the LINAC’s pulses to improve the signal-to-noise ratio, even with room lights on [115]. Published studies have demonstrated superiority to TLDs and OSLDs in obtaining clinically useful information during radiotherapy including the validation of surface dose to superficial targets, the verification of the spatial accuracy of treatment delivery in breast radiotherapy (such as unintended dose spillage to the contralateral breast), and the monitoring of field match lines during craniospinal irradiation and the breast and supraclavicular fields during breast and regional nodal treatments [116,117]. Additionally, Cherenkov imaging has niche applications, such as quality assurance for MR-LINAC systems [118], real-time imaging during CyberKnife radiotherapy [113], and quality and delivery control of preclinical FLASH radiotherapy [119,120]. While Cherenkov imaging is limited to surface interactions and cannot provide information about deeper volumes, it offers unique real-time tracking capabilities in non-proton radiotherapy. This feature will be particularly beneficial as radiation oncology shifts toward hypofractionation and FLASH radiotherapy, which deliver higher doses of radiation in shorter timeframes.

## 5. Charged Particle and Proton Therapy

Charged particle radiotherapy (CPRT) is a form of external beam RT that uses energetic charged particles, such as protons and carbon ions, to treat cancer. The most common form of CPRT is proton radiotherapy (PRT), which originated in the mid-20th century but has gained significant attention over the past two decades [121]. Compared to conventional photon radiotherapy, it can offer reduced normal tissue dose and therefore allow tumor dose escalation [122]. However, the financial cost of CPRT delivery and maintenance is much higher than any photon radiotherapy technique, becoming the major barrier to its widespread clinical adoption [123].

### 5.1. Physics Principles and Advantages

A key feature of CPRT is the Bragg peak, where the majority of the charged particle’s energy is deposited at a specific depth, beyond which there is minimal radiation. The Bragg peak position is determined by beam energy for a given particle. Thus, by changing the proton beam energy in proton therapy, the depth of energy deposition can be controlled. This allows for more precise targeting of tumors with less damage to surrounding normal tissues, potentially leading to fewer side effects compared to conventional photon radiotherapy. CPRT is especially beneficial for pediatric patients, as children are more sensitive to radiation and the reduction in long-term risks of radiation-induced secondary cancers is even more desirable.

Relative biological effect (RBE) is another critical concept of CPRT. RBE compares the biological effectiveness of charged particle radiation to X-ray radiation. CPRT is generally more effective than conventional RT due to a higher RBE. For instance, the average RBE of a proton beam at the midpoint of a spread-out Bragg peak (SOPB) in vivo is around 1.1, which is commonly used in current proton planning [124].

### 5.2. Challenges

The high cost and complexity of CPRT technology pose significant challenges. PRT or CPRT centers are expensive to build and maintain, and the cost-effectiveness of these treatments remains controversial compared to conventional radiotherapy. In the United States, despite the recent increase in charged particle treatment facilities, only 45 proton/heavy charged particle treatment centers are operational [125]. The limited availability is due to both the high cost and complexity of the technology. Limited availability, coupled with varying insurance coverage across countries and insurers, restricts patient access to these advanced treatments. Some insurers still consider proton therapy experimental for many indications [126,127].

Double scattering proton therapy (DSPT) and pencil beam scanning proton therapy (PBSPT) are two major proton treatment techniques currently used in clinical practice. Compared to DSPT, PBSPT is a notable advancement that allows for better dose conformity and more precise targeting of tumors through intensity modulation. Proton therapy is commonly used for pediatric cancers, central nervous system tumors, re-irradiation, and cancers of the head and neck, prostate, and breast. Numerous clinical trials have compared its efficacy to conventional photon therapy [128,129,130]. While plan comparisons usually confer a dosimetric advantage of proton and heavy-charged particle therapy over photon therapy, it does not always translate into an improved clinical outcome. For example, some recent studies showed that DSPT leads to similar clinical outcomes, compared to intensity-modulated photon therapy [129]. The lack of clinical evidence for the dosimetric benefits promised by charged particle therapy in these examples may be manifold: lack of delivery sophistication and primitive dose calculation, image guidance, and handling other treatment planning, targeting, and delivery uncertainties in these early trials when compared with mature photon RT. To this end, newer techniques such as PBSPT appear to be promising to better translate physics into clinical benefits, but ongoing research is needed to confirm that and to determine the types of cancers for which proton therapy is more efficacious than lower-cost photon RT [131].

### 5.3. Present and Future Developments

Technological advancements have led to the development of more compact proton therapy systems, from multi-gantry systems to single-gantry systems and, more recently, upright systems. For example, the MEVION S250-FIT therapy system fits within a standard photon LINAC vault. By eliminating the rotating gantry and using an upright robotic couch/chair system to position patients vertically, these compact systems reduce the footprint, cost, and complexity of proton therapy [132].

Proton arc therapy is another advanced PRT technique, which has the potential to further improve treatment plan quality with additional degrees of freedom, increasing dose conformity and decreasing normal tissue dose [133].

Dose calculation methods have also evolved. While early proton therapy relied on pencil beam algorithms, Monte Carlo dose calculations, which offer improved accuracy, are now more widely used. Range uncertainty, a significant challenge in proton and heavy charged particle therapy, is increasingly addressed through spectral tissue decomposition using multi-energy CTs. Cutting-edge plan optimization algorithms now incorporate variable linear energy transfer (LET) rather than assuming a nominal RBE, enabling more accurate dose optimization.

Image-guided radiation therapy with photons introduced sophisticated imaging and real-time guidance technologies to ensure the accurate delivery of planned radiation to tumors [134,135]. Recently, there have also been more advancements in integrating imaging guidance with proton therapy to combine the precision of proton therapy with the benefit of IGRT, compared with the early days of proton therapy, and the effort is still ongoing. Besides the conventional imaging techniques such as kV planar X-ray, CBCT, CT on rails, MRI, and surface tracking, proton portal imaging is also emerging as a potential alternative for patient positioning verification.

Additionally, by combining CPRT with other advanced techniques such as FLASH, surface guided RT, and adaptive RT, the treatment outcome of CPRT may be further improved, reducing healthy tissue damage or increasing tumoricidal effects on radio-resistant tumors.

On the clinical implementation side, increasing evidence supports the efficacy and safety of proton therapy, particularly for pediatric cancers, head and neck cancers, and certain types of brain tumors. However, large-scale randomized trials are still needed to compare long-term outcomes directly with conventional radiotherapy. Proton therapy’s impact on the immune system is another important subject of ongoing research, particularly in how it can be harnessed to improve cancer treatment outcomes when combined with immunotherapy.

Heavy charged particle therapy, such as carbon ion radiotherapy, offers even sharper Bragg peaks, reduced lateral scatter, and higher LET and RBE compared to proton therapy, potentially achieving superior dosimetry [136]. However, the technology requirements and costs are even higher for such systems. Currently, only one carbon ion therapy facility is under development in the United States, while a dozen or so are operational or in development worldwide [137]. Clinical trials comparing carbon ion therapy, proton therapy, and the conventional RT approaches will be needed to assess its cost-effectiveness and suitable types of diseases to treat.

Charged particle radiotherapy represents a significant advancement in cancer treatment, offering precise tumor targeting and the potential for reduced side effects. Ongoing improvements in optimization, imaging, and delivery techniques are enhancing the precision and effectiveness of CPRT. While challenges such as cost and accessibility remain, continued research and technological advancements are likely to expand its use, benefiting a broader range of patients in the future.

## 6. Adaptive Radiotherapy

Adaptive radiotherapy (ART), driven by the recent development of integrated systems capable of delivering online adaptation using advanced imaging modalities, is currently a major focus of clinical investigation, implementation, and development in RT [138].

The conventional paradigm of radiotherapy relies on anatomical images acquired at a single timepoint during treatment simulation. However, significant anatomical changes can occur during the treatment course, necessitating adjustments to the treatment plan to maintain accuracy [139,140]. ART leverages advances in imaging and AI to modify treatment plans based on updated anatomical data. Traditional radiotherapy employs margins to account for positional, delivery, and anatomical uncertainties [141]. Over the past two decades, technological advancements have significantly improved positional and delivery accuracy, driving a trend toward margin reduction, dose conformality, and dose escalation [142,143,144,145,146,147,148]. However, anatomical changes—such as intra-fractional organ movement, tumor shrinkage, and patient weight changes—have emerged as limiting factors for treatment accuracy and precision, compromising effectiveness. ART aims to counteract these challenges [149,150,151,152,153].

### 6.1. Adaptive Workflow

In terms of adaptive workflow, ART can be categorized into two main types: offline and online. Offline ART, which has been used clinically for many years, involves re-simulating and re-planning partway through a radiotherapy course to account for changes deemed significant by the attending oncologist. In this workflow, plan adaptation occurs offline between sessions, based on updated anatomy at a single timepoint, and is applied to future treatment sessions. In contrast, online ART conducts the entire workflow within the same session, developing a new plan based on the session’s anatomy and applying it immediately for treatment. Several integrated systems have recently become available to facilitate online ART workflows [81,95,154,155]. Although these systems are more complex and resource-intensive, online ART provides a more up-to-date treatment plan that accounts for anatomical changes occurring at each treatment fraction. A third, future ART workflow—real-time ART—remains under development. In this approach, plan adaptation would occur instantaneously, rather than over tens of minutes as in online ART, enabling real-time adaptation to anatomical changes. Figure 3 illustrates the differences between conventional RT, offline ART, online ART, and real-time ART.

The increasing adoption of online ART has been enabled by workflow automation, advanced imaging, and AI. ART is particularly beneficial for cancers such as cervical, prostate, bladder, pancreatic, head and neck, lung, and breast cancers [156,157,158,159,160,161]. When combined with margin reduction and/or dose escalation, ART can optimize treatment accuracy and efficacy while reducing side effects. Numerous studies have demonstrated the feasibility of ART across various systems and cancer types, highlighting its dosimetric benefits [156,157,158,159,160,161]. These studies indicate that adapting treatment plans to patient anatomy changes improves target volume coverage, spares normal tissues, and enhances clinical outcomes. Current research focuses on clinical trials to investigate the benefits of ART, as well as efforts to optimize technology, techniques, workflows, and the frequency of ART for different cancers [162].

### 6.2. Imaging

Advanced imaging modalities form the foundation of current ART platforms. MRI-based RT platforms, with their superior soft-tissue contrast and real-time, non-ionizing radiation imaging, were among the first to enable online ART [81,154,155]. Initially introduced as image-guided radiotherapy (IGRT) systems, these platforms later integrated ART workflows. MRI-based ART systems are widely used for gastrointestinal and genitourinary cancers and offer potential for biologically guided ART, as well as multi-parametric imaging for treatment monitoring, prognosis prediction, and personalized treatment stratification. Since its introduction, the CBCT-based ART platform has gained popularity due to its streamlined workflow, which enables efficient ART sessions lasting 15–30 min, and its lower cost compared to other systems, facilitating widespread adoption. The new HyperSight™ CBCT (Varian Medical Systems, Palo Alto, CA, USA) and upcoming ART workflows are expected to further improve CBCT image quality, allowing adaptive planning directly on CBCT images without the need for synthetic computed tomography (CT) [95,162,163]. More recently, a C-arm linac CBCT-based ART platform has also become available. The newest addition to the field is the PET-based platform, which is pending regulatory approval for online ART [164,165]. This system is poised to enable biologically guided ART and leverage further developments in tracers to synergize with other biological advancements.

### 6.3. Automation

Automation of key RT workflow steps, such as segmentation and plan optimization, has been instrumental in enabling online ART. AI techniques are expected to further improve the accuracy and robustness of these automated processes. However, current automation still requires significant manual verification and editing, making online ART resource-intensive. The implementation of ART faces several challenges, including the need for specialized equipment, training, and integration of AI technologies. Resource considerations are critical, as ART, particularly online ART, demands considerable time and expertise. Ongoing research focuses on optimizing workflows, reducing resource demands, and enhancing patient outcomes. Despite these challenges, ART represents a significant advancement in personalized cancer treatment, offering the potential for more tightly conformal and highly personalized radiation treatments that adapt to changes in patient anatomy and biology.

### 6.4. Current Trials and Early Clinical Evidence

Early clinical investigation of ART began with offline adaptation as well as earlier plan library approaches, which helped lay the groundwork for current online ART strategies. Trials such as ARTFORCE for head and neck cancer and RTOG 1106 for non-small cell lung cancer often used mid-treatment replanning based on anatomical or functional imaging such as FDG-PET or repeat CT scans to individualize dose escalation or dose re-distribution [166,167]. The RAIDER trial for bladder cancer used a plan library approach with plans based on varying sizes of bladder to be selected at treatment to best match bladder sizes and restore dose distribution on changed anatomy [168]. These studies demonstrated the technical feasibility and safety of ART, though consistent improvements in clinical endpoints such as survival or local control have not yet been uniformly observed. Building on this foundation, recent trials have increasingly shifted toward online ART. Two key examples are the SMART and MIRAGE trials. SMART, multi-institutional studies in pancreatic cancer and other challenging-to-treat malignancies in the abdomen or central thorax, used daily MR-guided online ART to safely deliver ablative doses, with minimal toxicity and promising early survival outcomes [84,169,170]. MIRAGE, a randomized phase III trial in prostate cancer, demonstrated that MR-guided ART significantly reduced acute and late genitourinary and gastrointestinal toxicity compared to CT-guided non-adaptive SBRT, while maintaining equivalent tumor control [171,172]. In addition to MR-guided platforms, multiple ongoing trials are now evaluating CBCT-guided online ART, particularly in pelvic and thoracic cancers, though outcome data from these studies are still maturing. With the booming adoption of online ART, exciting new clinical results are expected in the near future.

## 7. Hyperthermia

Heat has been used in cancer treatment since before the advent of radiation therapy. In the late 19th century, W.B. Coley induced fever in patients with inoperable sarcomas using what became known as “Coley’s toxin”, reported in ten cases [173]. Hyperthermia (HT), the targeted heating of tissue to approximately 40 °C using various modalities, remains an active area of cancer therapy research today [174]. This review focuses on the synergistic combination of hyperthermia and radiation therapy, first described in preclinical studies during the 1970s [175]. Thermoradiotherapy (TRT) has demonstrated improved outcomes over radiation therapy alone in a variety of tumor sites, including the breast [176], cervix [177], esophagus [178], head and neck [179], and sarcomas [180].

In contemporary radiation oncology, TRT is receiving renewed interest, particularly in combination with emerging therapies such as immunotherapy [181] and nanotechnology. The potential for HT to synergize not only with radiation but also with these modern modalities could expand its clinical utility beyond the currently established indications. This review first discusses the radiobiological synergy of TRT and its current clinical applications, followed by an exploration of future directions.

### 7.1. Biological Basis of Thermoradiotherapy

Radiation therapy primarily exerts its effects through DNA damage, either via direct ionization or the generation of reactive oxygen species through the radiolysis of water, ultimately leading to DNA oxidation [182]. The resulting single- and double-stranded DNA breaks, if unrepaired, can halt the cell cycle and trigger cell death. Hyperthermia enhances radiation-induced cytotoxicity by impairing the function of proteins involved in DNA repair when tissue temperatures exceed 40 °C [183]. Additionally, HT reduces the reproductive capacity of tumor cells, thereby increasing radiosensitivity [184].

Another synergistic mechanism involves the transient vasodilation of tumor vasculature caused by HT, which enhances perfusion. This is particularly important because tumor hypoxia—common in the core of larger tumors—limits the efficacy of radiation by reducing reactive oxygen species formation. By improving oxygenation, HT mitigates hypoxia-induced radioresistance. Moreover, HT has been investigated as a means to reduce radiation dose during re-irradiation for recurrent disease. For instance, in the treatment of locally recurrent breast cancer, a hypofractionated regimen (5 × 4 Gy, one fraction per week) achieved comparable complete response rates to those reported with higher mean doses [185].

### 7.2. Clinical Implementation

Currently, HT is delivered through microwave or radiofrequency-based systems [186]. Local, regional, or whole-body heating strategies have been explored [187], although recent findings suggest that local or regional HT may suffice even for metastatic disease via the induction of an abscopal effect [188]. Outstanding questions remain regarding the optimal temperature for radiosensitization [189], the precise molecular mechanisms of DNA repair inhibition [190], the ideal timing and sequencing of HT and radiation [191], and the accurate measurement of intratumoral temperatures.

Technological advances such as MRI thermometry have enhanced the ability to optimize temperature distribution and avoid hot spots. These developments may facilitate integration with novel techniques, including magnetic nanoparticle hyperthermia and immunotherapy. Magnetic nanoparticles, when exposed to an alternating magnetic field, can generate heat. Due to the increased permeability of tumor vasculature, these particles can accumulate preferentially in tumors following intravenous injection. This targeted heating, followed by radiation therapy, is currently under clinical investigation in glioblastoma [192] and prostate cancer [193]. This strategy may allow for more selective heating, enabling higher intratumoral temperatures while sparing surrounding normal tissues [194].

### 7.3. Challenges and Future Directions

Despite its promise, TRT remains a niche modality in most radiation oncology centers. For broader clinical adoption, robust data from modern clinical trials are needed to demonstrate TRT’s benefits across more cancer types. Much of the existing evidence was generated between 1980 and the early 2000s—an era prior to major advances in both radiation therapy and hyperthermia technology. New studies employing state-of-the-art techniques are essential not only to improve the evidence base but also to address reimbursement barriers in various healthcare systems. Additionally, clinical data on the integration of HT with particle therapy are sparse. Future trials should explore the potential advantages of combining hyperthermia with this advanced radiation modality.

## 8. Theranostics

In the era of personalized medicine, theranostics has gained significant interest over the last two decades. It refers to radiopharmaceuticals that combine therapeutic and diagnostic capabilities, typically imaged with PET/CT or PET/MRI. The radionuclide is bound to a ligand with high specificity to tumor receptors or biological pathways [195,196], thereby delivering a high radiation dose to the tumor microenvironment while sparing healthy tissue and minimizing toxicity [197]. Radionuclides can emit γ-rays, which are detected by imaging systems such as SPECT, PET, and scintillators. Other theranostic radionuclides emit α or β particles for therapeutic applications.

The therapeutic properties of these particles are governed by their LET and range. α-particles have a high LET (~80 keV/μm) and a short range (50–100 μm), making them particularly beneficial for treating targets near critical organs. In contrast, β-particles have a lower LET (~0.2 keV/μm) and a longer range (≤12 mm), making them more suitable for treating larger tumors. However, β-particles rely on oxygen to generate free radicals for single-strand DNA breaks, which makes them less effective in hypoxic tumors. In comparison, α-particles induce DNA double-strand breaks and are not oxygen-dependent. Examples of β-emitters include ^177^Lu, ^68^Ga, and ^131^I, while ^212^Pb is an example of an α-emitter.

### 8.1. Internal Dosimetry

Due to the heterogeneous tumor microenvironment and various washout mechanisms, personalizing radiopharmaceutical treatment remains a challenge. A commonly used clinical metric to quantify radioactive uptake is the standardized uptake value (SUV), which represents the uptake in a region of interest normalized by body weight. Guidelines such as PERCIST 1.0 (PET Response Criteria in Solid Tumors) [198] and those from the European Organization for Research and Treatment of Cancer [199,200,201,202,203] provide criteria for assessing treatment response using SUV published guidelines to assess the treatment response with SUV. However, SUV is a lumped constant subject to inter-observer variability, body composition differences, and image noise [204]. It is utilized in various commercial software applications, including MIM (MIM Software), XD4 (Mirada Medical), Syngo.via (Siemens), PET VICAR (GE Healthcare), and Advantage Workstation (GE Healthcare).

For more accurate personalized dosimetry, the Committee on Medical Internal Radiation Dose (MIRD) introduced the S factor, estimated using Monte Carlo simulations to account for radiation type, energy, source and target organ locations, and the relative geometry of the region of interest (ROI) [205]. This factor is multiplied by the time integral of the time-activity curve—obtained from multiple timepoints post-injection—to estimate the absorbed dose in the ROI. The S factor is incorporated into commercial dosimetry software, including MIM SurePlan MRT (MIM Software), OLINDA/EXM, and MIRDcalc [206].

Another widely accepted quantification method in non-clinical and research domains is pharmacokinetic modeling, which provides physiological insights into tissue permeability, blood flow, and tracer uptake, which can provide information needed for optimal dosing. Although primarily used for imaging tracers, the same principles can be applied to therapeutic tracers to optimize dosing strategies [207,208,209].

### 8.2. Current and Future Perspective of Theranostics

Theranostics began in the 1940s with the use of radioiodine, ^131^I-NaI, for the treatment of malignant and benign thyroid diseases. Since then, several theranostic agents have received FDA approval. A list of theranostic agents currently in clinical use is provided in Table 1, while agents under clinical investigation (as of July 2024) are listed in Table 2 (clinicaltrials.gov).

Following the success of theranostic agents targeting PSMA ligands, research is now exploring these ligands for other cancer types. Since PSMA is highly expressed in certain non-prostate tissues, it is being investigated for potential treatments in renal cancer, gastrointestinal cancers, sarcomas, and brain tumors. Additionally, PSMA-based theranostics are being explored as combination therapies with stereotactic ablative body radiotherapy (SABR) for oligometastatic disease.

Ongoing clinical trials are also investigating α-emitting radiopharmaceuticals, which have high biological effectiveness. However, since α-emitting agents are relatively new, most clinical trials remain in phase I/II, primarily focusing on safety assessments. As research progresses, theranostics is expected to further expand, integrating novel ligands and imaging strategies to enhance personalized cancer treatment.

## 9. Artificial Intelligence and Data Science

Artificial intelligence and big data science have emerged as transformative forces in healthcare research and clinical implementation. As a specialty rooted in advanced technology and medical imaging, radiation oncology is at the forefront of these efforts, leveraging AI to improve precision, efficiency, and patient outcomes [213]. The integration of AI in radiation oncology spans a wide range of applications, including image generation and segmentation, treatment planning, outcome and toxicity prediction, patient management, and quality assurance, all of which contribute to enhancing the quality of cancer care [214].

### 9.1. Image Generation, Enhancement, and Registration

AI techniques are increasingly integrated into imaging devices used in radiation oncology to improve image quality and reduce artifacts [215,216]. Deep learning (DL) models, such as convolutional neural networks (CNNs), have been successfully employed to mitigate common imaging artifacts, including metal and scatter artifacts in CT images and motion artifacts in MRIs [217,218,219,220]. AI also plays a critical role in reducing imaging noise and dose while improving the quality of volumetric image guidance modalities. These advancements are particularly important for adaptive assessment, treatment planning, and treatment-image-based outcome prediction. For example, AI has significantly enhanced the quality of CBCT images. Additionally, methods like generative adversarial networks (GANs) have been used to generate synthetic CT images from alternative sources such as CBCT, MRI, or surface imaging, enabling MRI-only simulations and CBCT- or MRI-based online adaptive therapy [221,222]. Conversely, CT has been used to generate synthetic MRI images, improving workflows in brachytherapy [223]. Another key area of AI application is image registration, where techniques like CNNs and GANs enhance the performance of deformable registration or augment data for registration training. Reinforcement learning has also been applied to evaluate and optimize a series of transformations for achieving optimal registration [214,224,225]. These AI-driven image registration methods improve delineation accuracy in treatment planning and enable the integration of multi-modal images, providing comprehensive information about the disease.

### 9.2. Image Segmentation

Segmentation of targets and organs at risk is a critical step in radiotherapy. Traditionally, this process requires significant effort from highly trained professionals, including radiation oncologists, medical dosimetrists, and medical physicists, and is prone to inter- and intra-observer variability. AI research and implementation have been particularly active in this area, aiming to automate the labor-intensive task of delineating tumors and normal tissues, which is essential for accurate treatment planning. DL-based segmentation methods have shown great promise in automating this process and improving its consistency [226]. Commercial AI tools for auto-segmentation are increasingly being adopted in clinical practice, with numerous reports highlighting gains in efficiency, consistency, and quality [165,227,228].

### 9.3. Treatment Planning

AI also plays a pivotal role in automating and optimizing treatment planning in radiation oncology. Traditional treatment planning is a manual, iterative process that involves collaboration among experts to create individualized plans based on simulation images. This process is often time-consuming and labor-intensive, leading to variations between planners and institutions. Knowledge-based planning (KBP) and advanced AI methods have been widely studied and implemented to improve and automate treatment planning [229]. KBP leverages data from previous high-quality plans to infer relationships between anatomy and dose, enabling the creation of efficient and consistent treatment plans for new patients. Advanced AI methods, such as 3D dose prediction using CNNs, GANs, and reinforcement learning, provide accurate spatial dose distributions, enhancing plan quality and efficiency while enabling online adaptive radiotherapy [164].

### 9.4. Quality Assurance

Quality assurance (QA) is essential in radiation oncology to ensure the safety and effectiveness of treatments. AI is increasingly being used to enhance QA processes, both for equipment and patient-specific QA [230,231]. AI models can predict potential machine failures and discrepancies in treatment delivery, suggesting preventative maintenance measures to prevent interruptions and ensure accurate dose administration. AI-driven tools also assist in detecting and correcting contouring errors and other anomalies during chart reviews and audit processes. These tools reduce manual workloads, increase the reliability of treatment plans, improve consistency in reviews, and enhance the accuracy of treatment delivery.

### 9.5. Outcome and Toxicity Prediction, Patient Management

Predicting patient outcomes and potential toxicities from radiotherapy is another area where AI has demonstrated significant promise [232,233,234]. By analyzing large datasets of clinical and imaging information—both before and during treatment—AI models can identify features and correlations that predict adverse effects, such as radiation-induced toxicity and treatment response. These predictions enable clinicians to strategize among treatment options and tailor therapies to individual patients, minimizing risks and improving overall outcomes. For example, AI algorithms have been used to predict radiation pneumonitis in lung radiotherapy based on patient-specific factors and dose parameters [235]. Other applications include AI-based imaging and clinical and biological marker assessments to personalize radiotherapy prescriptions, such as dose escalation or de-escalation [236].

### 9.6. Other Applications

Beyond the applications summarized above, AI and data science are being utilized in numerous other areas within radiation oncology. These include DL-based prediction of more accurate radiation doses from less precise algorithms to increase speed and accessibility [237], large language models for data labeling and extraction to support large-scale data curation and generative medical information [238,239], and many more. AI and data science are also driving efforts to integrate multi-modal and multi-scale data across radiation oncology, immunology, medical oncology, and other biological fields. These integrations are anticipated to lead to more personalized, efficient, and effective cancer care [240].

### 9.7. Current Challenges and Future Outlook

Despite these promising advancements, integrating AI into clinical workflows presents several challenges. Technical issues include data standardization and the generalizability of AI models, while the “black box” nature of many AI systems raises concerns about transparency and interpretability. Ethical considerations, such as data privacy and bias, also need to be addressed. Efforts are underway to develop “explainable” AI models that provide transparency in decision-making processes, as well as to establish regulations ensuring data diversity and model fairness [241,242,243]. Equally important are the challenges of validation and regulatory approval. Many AI models perform well in retrospective or single-institution data but lack the rigorous prospective, multi-center validation needed for clinical credibility [244,245]. The absence of standardized benchmarks also poses significant challenges for large-scale validation, placing much of the burden on individual institutions to validate AI tools locally at the time of deployment. At the same time, while some efforts are starting, regulatory frameworks tailored to medical AI are still underdeveloped compared with those for conventional medical devices and drugs, further complicating deployment [246,247,248,249]. Regulatory bodies are still evolving approaches to evaluate safety, efficacy, and post-market surveillance of AI-driven tools. Addressing these challenges is crucial for the successful adoption of AI in clinical practice.

On the deployment side, incorporating AI education and training into the curricula of radiation oncologists, medical physicists, and other healthcare professionals is essential. Continuous learning opportunities, such as vendor-provided training, academic courses, and workshops, are critical to equipping the current and future workforce with the skills needed to implement and interpret AI tools effectively

## 10. Radioimmunology

Although RT is a localized treatment, it plays a crucial role in modulating immune responses, potentially inducing systemic, immune-mediated anti-tumor effects. Radiation can directly stimulate or suppress immune responses, induce immunogenic cell death, modify the tumor microenvironment, and alter the tumor’s immunological profile. Immune cells such as phagocytes, B cells, and T cells can respond to radiation-induced DNA damage. B cells exposed to DNA can present antigens and release cytokines, further activating other immune cells. Phagocytes assist in clearing cellular debris, while DNA exposure to T cells can promote the production of type 1 interferons and enhance tumor infiltration. Increased T cell penetration into tumors, driven by chemotactic factors released from exposed DNA and mitochondrial genomes, enhances the immunological response to radiation. However, prolonged DNA radiation exposure can also induce T cell apoptosis, reducing their anti-tumor efficacy [250].

Figure 4 illustrates the major immune pathways modulated by radiation therapy, including both stimulatory and suppressive effects within the tumor microenvironment and systemic immune responses. It highlights how radiation-induced DNA damage and cellular stress lead to immune cell activation, cytokine release, and potential abscopal effects.

Radiotherapy to a primary tumor at one site has been observed to occasionally result in an abscopal effect, which is the recession of metastatic cancer at other sites [251]. This phenomenon is thought to be mediated by the immune response triggered by radiation at the primary site, which also targets distant tumor cells. The generation of reactive oxygen species and DNA breaks caused by radiation, along with the subsequent biological processes, are essential for these immunomodulatory effects.

### 10.1. Effect of Radiation on the Immune System

Ionizing radiation can act as both an immunostimulant and an immunosuppressant. Immunosuppression may contribute to tumor recurrence, while immunostimulation is driven by mechanisms such as the upregulation of interferons, the release of epigenetically silenced tumor-associated antigens (TAAs), and the activation of damage-associated molecular patterns (DAMPs). These processes enhance antigen presentation by dendritic cells and increase the antigenicity of malignant cells through interferon beta 1 [252].

However, radiation can also exert immunosuppressive effects, particularly in the tumor microenvironment. These include the upregulation of programmed death-ligand 1 (PD-L1), which inhibits activated T cells and natural killer (NK) cells, and the secretion of transforming growth factor beta 1, which promotes DNA damage repair and reduces tumor radiosensitivity. Radiation can also recruit tumor-associated macrophages, tumor-associated neutrophils, and regulatory T cells (Tregs), which can promote tumor growth, angiogenesis, and metastasis [253,254,255].

### 10.2. Radioimmunotherapy and Radiation Abscopal Effect

The immunosuppressive effects of RT can be mitigated by combining RT with immunotherapy. One of the most studied approaches is the use of immune checkpoint inhibitors, which block immunosuppressive mechanisms. Hypofractionated RT, for example, has been shown to effectively promote the abscopal effect while minimizing the stimulation of immunosuppressive cells like Tregs, macrophages, and myeloid-derived suppressor cells [251,256]. Clinical trials, such as those combining stereotactic body radiotherapy with ipilimumab (NCT02239900), have demonstrated greater clinical benefits with sequential rather than concurrent therapy [257]. Recent advances in radioimmunotherapy have shown particular promise in hematologic malignancies. Preclinical and clinical studies have demonstrated the efficacy of radiolabeled monoclonal antibodies in treating non-Hodgkin’s lymphoma, providing targeted delivery of radiation while simultaneously engaging immune-mediated cytotoxicity [258]. These strategies exploit tumor-specific antigens to enhance selectivity and minimize off-target effects. Furthermore, randomized clinical trials assessing the use of radionuclide/monoclonal antibody conjugates have highlighted their potential to improve progression-free survival and overall response rates in various solid tumors, particularly when used in combination with external beam radiation [259]. These findings underscore the growing role of molecularly targeted radioimmunotherapeutics in bridging localized RT and systemic immune modulation.

While the abscopal effect with RT alone is rare and dose-dependent, combining RT with immunotherapy enhances this effect microscopically. However, the optimal timing and dosing of immunotherapy relative to RT remain unclear. Radiation at primary tumor sites can increase plasma concentrations of chemokines and cytokines, induce delayed DNA damage responses in distant tissues, and alter the tumor microenvironment at abscopal sites. Tumor-derived exosomes at distant sites may also interact with immune cells, contributing to the abscopal effect [251,255,260]. Several checkpoint inhibitors combined with RT have shown improved progression-free survival and enhanced abscopal effects, leading to FDA approval and widespread clinical use (Table 3).

### 10.3. Future Directions in Radioimmunology

High-dose radiotherapy (H-XRT) has demonstrated benefits such as reversing resistance to anti-PD1 therapy, promoting T-cell priming, and releasing proinflammatory mediators like DAMPs. However, it can also upregulate immunosuppressive cells like Tregs and macrophages [251,256,261]. To amplify the abscopal effect, researchers are exploring optimal sequencing, dosing, fractionation, and radiation sites. One promising approach is the radscopal effect, which uses low-dose RT (0.5–2 Gy per fraction, totaling 1–10 Gy) in combination with immunotherapy to reprogram the tumor microenvironment, making it more receptive to immunotherapy. In this approach, low-dose radiotherapy (L-XRT) is delivered to secondary tumors while high-dose radiation targets primary tumors. Combining L-XRT and H-XRT with checkpoint inhibitors is being investigated in clinical trials (e.g., NCT05039632) and has shown potential to enhance immune cell infiltration, tumor killing, and abscopal responses [255,261].

### 10.4. Radiolabeled Immunotherapy

Another branch of radioimmunology called radiolabeled immunotherapy (RIT) involves the use of radiolabeled antibodies to molecularly target tumor cells. This approach, like theranostics, exploits tumor immunology for precise targeting. RIT is highly effective in tumors with a uniform density of optimal cell surface antigens, provided the targeting antibodies are not expressed in normal tissues [262].

### 10.5. Challenges and Advances in RIT

Though RIT has great potential for cancer treatment, it is accompanied by the self-attacking of T cells in normal tissues. To enhance the radiation deposition in tumors while sparing normal tissues, various heavy metal nanoparticles like Au, Ag, Bi, and Gd are being explored as potential radiosensitizers and immune response stimulators [263]. Radionuclides commonly used in RIT like ^131^I have low penetrating power, poor targeting ability, and low radiation intensity. However, conjugating twin-arginine translocation peptide-modified nanoparticles with ^131^I have shown promise in enhancing the X-ray penetration depth by facilitating cellular uptake, infiltrating the subcellular tumor environment, and facilitating the entry of molecular cargo into the target. It could elicit an immune response through immune cell activation, intensifying phagocytosis, and tumoral infiltration [264]. Labeling nanoclusters with radionuclides like ^99m^Tc and ^177^Lu in mice has been shown to suppress distant metastatic tumor growth with long-term immune memory effects [265].

RIT utilizes monoclonal antibodies combined with cytotoxic radionuclides which selectively attach to the tumor-specific antigens like EGFR, Her2, or CD20. However, the efficacy and therapeutic index of RIT are limited in solid tumors due to its complex tumor microenvironment and hematopoietic toxicities. Radiosensitizers and pre-targeting approaches are currently being explored to overcome these barriers. Pre-targeting radioimmunotherapy (PRIT) can increase therapeutic indices and enhance imaging contrast by separating tumor targeting and radiotracer delivery [266,267]. Several PRIT investigations are in the preclinical stage or clinical trials, like ^177^Lu-DOTA (NCT05130255) and ^90^Y-IMP288 (NCT02300922). The effect of photon radiation on tumor cells is well established. However, the effects of particle therapies like proton and carbon therapy on tumor cells and various immune cells are less well understood. Due to the higher RBE and LET, it can induce stronger chromosomal aberrations and mutations than photons. Particle radiation in combination with immunotherapy could be a promising new combination therapy [268].

## 11. Spatially Fractionated Radiation Therapy

Spatially fractionated radiation therapy is an advanced radiation treatment technique, delivering an intentionally designed inhomogeneous dose to the target volume. SFRT originated from orthovoltage GRID therapy, which has been clinically applied since the 1930s [269]. Compared to conventional RT, SFRT was designed to significantly minimize radiation damage to normal tissues, including the skin, while still maintaining a comparable level of tumor control by delivering highly modulated radiation doses concentrated within the tumor area, effectively sparing surrounding healthy tissues. This approach has been particularly useful for treating bulky tumors (>5 cm) or inoperable lesions. Current clinical applications demonstrate that SFRT can achieve high rates of clinical response with reduced normal tissue toxicities. Recent radiobiological studies on cell signaling, vascular damage, and immune responses following radiation have provided further evidence supporting the successful outcomes of SFRT [270]. Figure 5 illustrates two SFRT techniques: GRID therapy and LATTICE therapy.

### 11.1. Prescription Parameters and Techniques

In contrast to conventional RT, the prescription of SFRT involves additional parameters: peak dose, valley dose, beam size (or vertex size), beam spacing (or space between vertices), and peak-to-valley dose ratio. As their names suggest, the peak dose refers to the higher dose delivered to specific regions, while the valley dose refers to the lower dose delivered to surrounding areas. The peak dose (typically ≥8 Gy) is generally used as the prescribed dose; however, treatment outcomes show a stronger correlation with the valley dose (usually <1/3 of the peak dose) than with the peak dose itself. Tumor response to SFRT also varies depending on the primary tumor site. For instance, Mohiuddin et al. demonstrated that squamous cell carcinoma (SCC) and soft-tissue sarcomas exhibit higher response rates to SFRT. In previous clinical trials, SFRT has been used both as a standalone treatment and in combination with conventional RT to treat various cancers, including melanoma, sarcoma, SCC, and others [271]. Several studies have shown that combining SFRT with conventional RT can improve tumor control rates, highlighting the potential of SFRT as a complementary approach to enhance treatment efficacy. Table 4 outlines common dosimetric parameters used in different SFRT techniques.

### 11.2. SFRT Techniques and Clinical Applications

SFRT can be applied using a variety of treatment techniques, including 2D RT, 3D conformal RT (3D CRT), IMRT, volumetric modulated arc therapy (VMAT), and proton RT. Based on dosimetric techniques, SFRT is primarily categorized into four subtypes: GRID therapy, LATTICE therapy, minibeam radiotherapy (MBRT), and microbeam radiotherapy (MRT). Table 5 summarizes the key similarities and differences between the 4 techniques.

**GRID Therapy**: This technique combines a brass or Cerrobend grid collimator with conventional planning techniques (2D RT or 3D CRT) [272,273,274]. Each field of a GRID therapy plan creates a matrix dose pattern of pencil-shaped beamlets [275].**LATTICE Therapy**: This approach utilizes advanced treatment techniques (IMRT, VMAT, pencil beam scanning PRT, arc PRT, etc.) to conform peak doses to contoured high-dose volumes (vertices) [276,277]. LATTICE therapy mimics the dosimetric design of GRID therapy but offers more convenient delivery without requiring a physical grid. It also provides flexibility for customized OAR sparing through beam modulation [278]. Figure 6 shows a simulated LATTICE plan.

The beam size (or vertex size) in GRID or LATTICE therapy is typically 1–2 cm, with beam spacing (or valley width) ranging from 1 to 4 cm. A significant number of patients have successfully undergone GRID or LATTICE therapy [279,280,281]. Table 6 summarizes selected clinical studies.

### 11.3. Minibeam and Microbeam Radiotherapy

MBRT operates on a millimeter or submillimeter scale (beam width: 0.3–1 mm; beam spacing: 1–4 mm), while MRT operates on a micrometer scale (beam width: 25–100 µm; beam spacing: 200–400 µm) [270]. The exploration of these techniques is motivated by the potential for higher selectivity in biological effects at cellular and microvascular scales, as well as more efficient dose utilization. Although MRT and MBRT have not yet been translated into clinical applications, numerous preclinical studies demonstrate their promising potential [282,283,284,285,286].

### 11.4. Radiobiological Mechanisms of SFRT

In addition to inducing DNA double-strand breaks, SFRT triggers or enhances multiple radiobiological mechanisms to achieve tumor control without delivering a uniformly high dose to the entire tumor region. The primary mechanisms proposed over the past few decades include radiation-induced cell signaling (bystander/abscopal effect), vascular damage, and anti-tumor immunity [270,271].

The cell signaling effect in SFRT is characterized by the phenomenon where tumor cells in the valley (low-dose) regions exhibit lower survival rates than expected based on the valley dose alone. This occurs due to signals—such as reactive oxygen and nitrogen species, cytokines, and other molecules—released by cells in the nearby peak (high-dose) regions [287,288,289,290].

Tumor vascular damage also plays a significant role in the therapeutic effect of SFRT. Tumor vessels are immature and more radiosensitive than normal vasculature. High doses (>10 Gy) can damage or kill vascular endothelial cells, leading to reduced blood perfusion and hypoxia in the tumor. This vascular damage can indirectly cause tumor cell death in the valley regions [291,292,293,294].

The immune effects of radiation are complex and have been widely studied [284,295,296]. Radiation can activate immune cells, including B cells, NK cells, CD8+ T cells, and CD4+ T cells, through various pathways, enhancing anti-tumor immunity [297]. It also triggers an inflammatory response, increasing the infiltration of immune cells into the tumor. Studies have shown a dose-dependent increase in inflammatory and immune responses within the dose range of 0.5–4 Gy [298]. However, high-dose RT (>5 Gy per fraction) can simultaneously suppress immune effects by recruiting immunosuppressive cells and enhancing immunoregulation. The inhomogeneous dose distribution of SFRT leverages both vascular damage and anti-tumor immune responses to improve tumor control [299].

### 11.5. Current Challenges and Future Directions

A major challenge in SFRT is the lack of consensus on treatment protocols, particularly regarding dosimetric parameters for different tumor sites. However, growing clinical interest has led to increased participation in preclinical and clinical SFRT studies. Recently, Nina A. Mayr et al. and Amendola, B.E. et al. developed multi-institutional consensus guidelines for SFRT clinical trial design [300,301]. Despite the availability of published consensus on clinical trial designs for certain treatment sites, designing a feasible clinical protocol remains challenging for radiation therapy teams with limited experience. For example, in the case of LATTICE therapy, further investigation is required to determine institution-specific parameters for SFRT, including structure contouring and plan quality evaluation.

The underlying rationale for SFRT is still not fully understood. Unlike conventional RT prescriptions, SFRT involves additional variables—such as beam size, spacing, and valley dose—beyond the standard considerations of target coverage and prescription dose. The relationship between SFRT prescription parameters and treatment outcomes has yet to be quantified. These uncertainties hinder the efficient and widespread adoption of SFRT in clinical practice.

Combining SFRT with other treatment modalities, such as immunotherapy and particle therapy, may further enhance treatment outcomes. For example, Lu et al. summarized current studies on the synergy between SFRT and immunotherapy [302]. Additionally, experts predict significant benefits from combining particle therapy with SFRT, and clinical trials for proton SFRT are underway (e.g., NCT06327477, NCT05121545).

Despite the lack of standardized protocols, current clinical case reports and ongoing trials highlight the significant potential of SFRT. With continued research and clinical efforts, SFRT is poised to become a routine treatment option, benefiting a broader range of patients.

## 12. FLASH Radiotherapy

FLASH radiotherapy is characterized by the ultra-fast delivery of RT at dose rates thousands of times higher than those conventionally used in clinical radiation oncology. While conventional radiotherapy dose rates are approximately 0.01 Gy/s, FLASH radiotherapy is broadly defined as employing dose rates greater than 40 Gy/s [303]. Achieving these ultra-high dose rates typically requires specialized treatment machines modified by removing specific beamline components to maximize dose output [304], or, in some cases, accelerators originally designed for industrial applications [305,306]. Currently, FLASH radiotherapy ultra-high dose rates have been achieved using photons [307], electrons [304], protons [308], and even particles heavier than protons like carbon ions [309].

### 12.1. The FLASH Effect: Historical Insights and Potential Biological Mechanisms

The FLASH effect in radiotherapy was first reported in 1959 by Dewey and Boag [310], who observed that delivering a dose of 10–20 kilorads in 2 microseconds induced hypoxic conditions in bacteria, thereby acting as a radioprotector compared to the same dose delivered at conventional dose rates. Since then, numerous preclinical studies have demonstrated the unique radiobiological effects of ultra-high-dose-rate radiation in mammalian [311] and eventually human cells [312]. The contemporary use of FLASH radiotherapy was furthered by results published by Favaudon [313] and authors in 2014. Using a live mouse model, they found that 100% of mice receiving 17 Gy to the lungs at conventional dose rates developed pneumonitis and fibrosis, whereas none of the mice treated with 17 Gy at FLASH dose rates exhibited these toxicities. Additionally, tumor control remained non-inferior between the two dose-rate regimens. Subsequent studies using various tumor and normal tissue models have shown similar findings of reduced radiation-induced toxicity at ultra-high dose rates [314,315,316,317]. Therefore, the FLASH effect is described as the effect of reduced normal tissue damages at the ultra-high dose rate as compared with the conventional dose rate, and FLASH radiotherapy is being actively researched often as a single-dose radiotherapy at ultra-high dose rates.

Even to this day, the complex biological mechanism of the FLASH effect in radiotherapy is not completely understood. The primary advantage of FLASH radiotherapy is the enhanced sparing of healthy tissues at therapeutic doses, but there is little consensus on how this effect is achieved. Expanding on Dewey and Boag’s work, studies suggest that ultra-high dose-rate irradiation can induce transient hypoxia, which acts as a radioprotectant for both normal and cancerous cells [316,318,319,320,321]. Additionally, some research indicates that hyperoxia negates the FLASH effect [305]. While no single radiobiological model fully explains the phenomenon, oxygen dynamics appear to play a critical role, with differences in the biochemical responses of normal and tumor cells contributing to the selective sparing of healthy tissues [322]. The rapid delivery of radiation leads to a phenomenon where tissue absorbs the energy at an accelerated pace, generating reactive oxygen species that interact with cellular components like DNA, leading to cell damage [323].

Other proposed mechanisms involve differences in oxygen depletion and reoxygenation kinetics between tumor and normal tissue environments, as well as potential immunological responses to vascular damage induced by FLASH dose rates. Several pre-clinical experiments have been conducted which give credence to this hypothesis. The ultra-high dose rate used during FLASH-RT is thought to modulate the inflammatory processes in the tumor microenvironment [324], perhaps enhancing anti-tumor effects [325,326,327].

### 12.2. Preclinical Studies, Early Clinical Results, Current Clinical Trials, and Future Directions

All of the promising preclinical work has since led to the transition of this technique to larger mammalian and human treatments. Prospective data collected during a trial [328] where thirteen canine cancer patients with superficial solid cancers or post-operative residual disease received RT at FLASH dose rates with an electron beam to evaluate the clinical efficacy of such a workflow. The authors of this study found that treatments in a clinical setting using FLASH radiotherapy were feasible, as partial or complete disease response was observed in most patients, and adverse effects were mild. The first published human case report [329] of FLASH radiotherapy described the treatment of a recurrent cutaneous T-cell lymphoma lesion. In this case, a single 15 Gy dose was delivered in 90 milliseconds using an electron beam, resulting in a complete response with minimal toxicity.

In order for FLASH radiotherapy to truly become an accepted and utilized treatment modality, clinical studies on humans are being proposed and carried out. FAST-01 [330,331] was a prospective nonrandomized clinical study to assess both workflow feasibility and clinical outcomes of proton FLASH radiotherapy treatments of patients for painful bone metastasis in the extremities. FAST-01 began accrual in 2020 and included 10 patients. The results of this study showed that it is feasible to deliver proton FLASH radiotherapy in a clinical setting, but that further studies and trials extending this treatment technique to other parts of the body are needed to demonstrate its applicability to multiple cancers. Additional studies are needed to prove the clinical utility, potentially based on the pain relief of the therapy. A good start to this is the FAST-02 study [331], which looks to assess toxicities and pain relief in ten patients using proton FLASH radiotherapy in patients with painful thoracic bone metastases. Data collected in prospective studies such as these will start to answer the question of whether FLASH radiotherapy can be delivered safely and effectively to humans, and eventually what clinical benefits are truly rendered from its use. Meanwhile, technical investigations and developments are underway to improve the dosimetry accuracy and monitoring at the FLASH dose rates, optimize the dose rate and modality for FLASH radiotherapy, and elucidate the underlying mechanism(s) of the FLASH effect.

## 13. Boron Neutron Capture Therapy

Boron Neutron Capture Therapy (BNCT) is not a recent innovation; it was first proposed as early as the 1930s [332,333]. However, after decades of limited activity following initial clinical trials in the 1950s, interest in BNCT has recently been reignited. This resurgence is driven by advancements in accelerator-based neutron sources, improvements in boron delivery agents, and the growing interest in theranostics.

The fundamental principle of BNCT is described by Equation (1): the isotope ^10^B absorbs thermal neutrons and decays into helium and lithium nuclei. Both He and Li have a limited range of a few micrometers and exhibit high linear energy transfer (LET). To achieve the therapeutic effects of BNCT, cancer-targeting boron delivery agents are used to concentrate ^10^B in cancer cells. The tumor site is then irradiated with low-energy thermal neutrons. When combined, boron agents and thermal neutrons produce a highly targeted cytotoxic effect, with the He and Li nuclei delivering lethal doses to cancer cells while sparing surrounding normal tissues, as a result of the He and Li nuclei having ranges on the same order as a typical cell diameter. While boron agents and thermal neutrons are relatively benign on their own, their combination in BNCT enables precise and localized cancer cell destruction. The efficacy of BNCT depends on two key factors: the ability to deliver a high concentration of ^10^B to cancer cells and the availability of a suitable neutron source for irradiation.(1)B510+n01→Li37+He24

BNCT was initially proposed for challenging cancers, such as high-grade gliomas, where conventional treatments have shown limited effectiveness. The first human treatment using BNCT took place at the Brookhaven National Laboratory in 1951 [334]. However, clinical trials in the United States were discontinued due to severe treatment toxicities, primarily caused by limited neutron penetration in deeper tumors and non-selective uptake of boron agents by normal tissues. High blood boron concentrations led to excessive toxicity in normal brain vessels and the scalp [335]. While BNCT investigations were stopped in the United States, Japan and several other countries in Asia and Europe continued to advance BNCT research [332,336,337,338]. Recent developments in accelerator-based neutron sources, boron delivery agents, and imaging technologies have propelled the recent resurgence in BNCT [339,340,341,342,343].

### 13.1. Neutron Sources

Unlike the nuclear reactor-based neutron sources used in early BNCT research, modern accelerator-based neutron sources, such as those utilizing cyclotrons and linear accelerators, have significantly improved the accessibility of neutron sources for BNCT investigations and clinical applications [339,344,345]. Collaborations with Japanese commercial companies have supported the development of advanced neutron sources with better monochromatic beams, higher and selectable energies, and increased intensities, all of which enhance the effectiveness of BNCT. Additionally, efforts are underway to make these neutron sources more compact and cost-effective, further facilitating their adoption in clinical settings.

### 13.2. Boron Delivery Agents

Boron delivery agents have evolved significantly since the early days of BNCT. First-generation agents, such as boric acid compounds, were followed by second-generation agents like sodium borocaptate and boronophenylalanine [346,347]. Today, third-generation boron delivery agents offer improved stability, selectivity, and specificity for tumor targeting. These agents include low-molecular-weight compounds, such as boron-containing amino acids, polyhedral boranes, biochemical precursors, DNA-binding agents, and boron-containing porphyrins, as well as high-molecular-weight agents, such as boronated monoclonal antibodies, boron-containing liposomes, receptor-targeting agents, and nanoparticles [348]. These advancements have enhanced the efficiency, concentration, specificity, and micro-distribution of boron delivery to tumor cells, including the ability to cross the blood–brain barrier [349].

### 13.3. BNCT as a Multimodal Therapy

BNCT combines the mechanisms of chemotherapy, targeted therapy, and radiotherapy. The success of BNCT depends on the ability to concentrate boron agents in tumor cells relative to blood and normal tissues. However, measuring boron concentrations in tissues is more challenging than in blood, and tumor heterogeneity—both within individual patients and across different patients—leads to variability in boron delivery and enhancement. To address this, researchers are exploring imaging techniques to measure boron concentrations, paving the way for theranostics in BNCT. PET/CT and MRI have been used to detect and image boron compounds, serving as surrogates for assessing BNCT effectiveness [350,351,352,353]. While this research is still in the preliminary stages, the success of ^131^I for thyroid theranostics [354] and the employment of newer approaches such as radiomics [355], theranostics research has the potential to improve the efficacy of BNCT.

### 13.4. Clinical Applications and Future Directions

BNCT has been applied to a variety of cancers, particularly those that are recurrent, refractory to conventional treatments, or difficult to target due to factors such as the blood–brain barrier. Clinical trials and research have explored BNCT for glioblastomas, intracranial tumor beds, melanoma, head and neck cancer, recurrent lung cancer, and recurrent meningioma, among others [336,356,357,358,359].

In a phase II trial for recurrent head and neck squamous cell carcinoma in Japan, accelerator-based BNCT with borofalan (^10^B) showed a 71% overall response rate and good one- and two-year survival rates [360]. The trial also showed low high-grade toxicity. These findings, which were validated in post-marketing surveillance including more than 160 patients, justified national regulatory approval. With median survival periods longer than historical controls, BNCT has also demonstrated promise in treating high-grade gliomas, particularly when paired with bevacizumab to treat radiation-induced edema [360]. Similar trials were also conducted in other regions. In spite of previous irradiation, a Finnish phase I/II study employing reactor-based BNCT for recurrent head and neck cancer in Europe found a 76% response rate, including 36% total remissions [361]. Although the results of European glioma studies employing borocaptate (BSH) were less impressive, they contributed to the development of safety standards and guided the shift to more efficient boron carriers, such as BPA [360,362]. Together, these investigations highlight BNCT’s potential to provide high-LET radiation that is tumor-selective with promising therapeutic results, particularly in situations where traditional therapies are not very effective. While the clinical application of BNCT remains limited, ongoing research is expected to lead to further breakthroughs, increase global accessibility, and standardize treatment protocols, ultimately benefiting more patients.

## 14. Other Areas of Interest

While the previous sections have focused on key advancements in RT, several promising areas of research are also shaping the future of radiotherapy. These innovations focus on improving therapeutic efficacy, minimizing side effects, and expanding treatment options for resistant or difficult-to-treat tumors. While still in experimental or early clinical stages, these approaches—including Auger therapy, hydrogen therapy, alpha-particle therapy, photodynamic therapy, and hyperthermia—offer promising new avenues in radiation oncology. The following sections provide an overview of these developments and their potential clinical applications.

### 14.1. Intraoperative Radiotherapy

While traditionally used in breast cancer treatment, Intraoperative Radiotherapy (IORT) is finding expanded applications. By delivering a high radiation dose directly to the tumor bed during surgery, IORT aims to minimize recurrence risk and protect surrounding healthy tissues. This technique demonstrates potential for improving survival rates and reducing post-operative complications. IORT is being investigated for head and neck cancer, pancreatic cancer, soft tissue sarcomas, and other solid tumors, particularly in cases of incomplete resection [363,364,365]. The INTRAGO and INTRAGO II clinical trials are exploring IORT’s efficacy in Glioblastoma [366].

### 14.2. VMAT Total Body and Total Marrow Irradiation

Volumetric Modulated Arc Therapy has become increasingly popular for total body irradiation (TBI). VMAT’s dose modulation capabilities enable precise dose conformity and manage heterogeneity in TBI, while minimizing organ toxicity to critical structures such as the lungs, kidneys, and brain [367,368]. For patients undergoing bone marrow transplantation, total marrow irradiation (TMI) offers an even more refined approach. TMI delivers escalated doses directly to the bone marrow, maximizing the destruction of diseased cells and improving treatment efficacy while reducing side effects [369].

### 14.3. Genomic Profiling

Analyzing tumor genetic makeup allows oncologists to predict patient responses to RT and personalize treatment. Recent advancements in genomic technologies enable the identification of specific biomarkers for risk stratification regarding disease progression and recurrence. This facilitates more precise and individualized treatment planning. Current research has provided insights into genomic profiling for prostate and lung cancers, among others. Looking forward, genomic profiling holds the potential to predict tumor and healthy tissue responses to radiation, enabling further personalization of treatment plans [370].

### 14.4. Three-Dimensional Printing

3D printing has revolutionized numerous aspects of radiation oncology [4,5]. Its applications include creating patient-specific boluses, compensators, immobilization devices, and phantoms for dose verification and quality assurance. This technology enables highly customized treatments, enhancing radiation delivery quality and reproducibility. For example, 3D-printed boluses can conform to irregular body surfaces like ears and noses, ensuring consistent dose distribution. Similarly, patient-specific compensators can be fabricated to personalize electron and brachytherapy treatments. Furthermore, 3D printing facilitates the development of anatomically accurate phantoms, essential for treatment planning and quality assurance. Notably, the PACER clinical trial is investigating the use of Electron Beam Intraoperative RT following chemoradiation in pancreatic cancer patients [371,372].

### 14.5. PTV Definition

The geometric concept of PTV margin from ICRU definition is becoming increasingly obsolete with the evolving optimization algorithms and dose calculation techniques in lung cancer radiotherapy, the development of proton and particle therapy, and the implementation of adaptive radiotherapy. Instead of relying on population-based geometric PTV margin expansions, the future paradigm will shift toward robust optimization that explicitly accounts for uncertainties, and adaptive planning that responds to daily anatomical and physiological changes. Ultimately, the PTV margin will be retired and replaced by a more biologically driven concept centered on spatially varying recurrence probability and patient-specific risk modeling. Such concept changes will mark a fundamental paradigm shift in how we define and target disease in radiotherapy.

### 14.6. Alpha-Particle Therapy

Alpha-particle therapy is a promising cancer treatment that utilizes highly energetic helium nuclei to destroy tumor cells while minimizing damage to surrounding healthy tissues. By intravenously injecting alpha-emitting radiopharmaceuticals, this approach leverages the high linear energy LET of alpha particles to deposit significant doses over short distances, effectively killing cancer cells. Alpha-emitter RT is already used to treat bone metastases from prostate cancer and is currently being investigated for other malignancies, including leukemia and melanoma. Ongoing research aims to improve alpha-emitter production and delivery mechanisms to enhance therapeutic efficacy and expand its clinical applications [373,374,375].

### 14.7. Photodynamic Therapy

Photodynamic therapy (PDT) is a clinically validated, minimally invasive treatment that selectively targets malignant cells with cytotoxic effects. The procedure involves administering a photosensitizing compound followed by exposure to light at a wavelength matching the sensitizer’s absorption spectrum. In the presence of oxygen, this interaction triggers a series of reactions leading to direct tumor cell destruction, microvascular damage, and a localized inflammatory response. PDT has demonstrated curative potential, particularly for early-stage cancers, and has been shown to extend survival in patients with inoperable tumors while preserving organ function. Its minimal toxicity, lack of systemic effects, and resistance to intrinsic or acquired resistance make PDT an attractive option for combination therapies. Recent technological advancements continue to position PDT as a valuable addition to standard cancer care [376].

### 14.8. Auger Therapy

Although first proposed over a century ago, Auger therapy has seen significant advancements in the last few decades. This technique utilizes radionuclides that undergo the Auger effect to emit low-energy electrons, which cause severe DNA damage when emitted within micrometers of the target. Auger therapy is particularly advantageous as it delivers lethal radiation doses to cancer cells while sparing nearby healthy tissues. Current research focuses on improving delivery mechanisms and enhancing the selectivity of Auger emitters to better target molecular structures within cancer cells, making it a promising option for treating radioresistant tumors [377,378].

### 14.9. Hydrogen Therapy

Hydrogen therapy is emerging as a potential radioprotective agent in radiation oncology. Although the exact mechanisms remain under investigation, hydrogen water and hydrogen gas have shown both anti-tumoral effects and the ability to alleviate side effects associated with conventional chemotherapeutics. These effects are believed to result from hydrogen’s antioxidant and anti-inflammatory properties. While still in the experimental stages, early studies suggest that hydrogen therapy could enhance the effectiveness and tolerability of radiation treatments, potentially improving patient outcomes [379,380].

## 15. Conclusions

We have reviewed a selection of exciting and novel approaches in radiotherapy for cancer that have emerged in recent years. Many of these technologies have recently been adopted clinically, significantly improving patient outcomes and expanding therapeutic options. Meanwhile, other promising advancements are still in development, with several undergoing clinical trials and awaiting full translation into standard practice. The major advantages, disadvantages, clinical readiness, and key challenges associated with each modality covered in detail and in the Other Areas of Interest section are presented in Table 7 and Table 8, respectively.

The successful integration of these innovations depends on rigorous clinical validation, multi-institutional collaborations, and strategic partnerships with industry leaders. Advancements in imaging, dosimetry, artificial intelligence, and precision medicine will further refine these techniques, ensuring their safety, efficacy, and accessibility. As regulatory approvals progress and reimbursement frameworks evolve, more patients will benefit from these cutting-edge therapies.

With continued research and investment, the field of radiation oncology is poised for a transformative shift, where novel therapies become routine components of cancer care. By fostering collaboration and evidence-based implementation, we can look forward to a future in which precision radiation treatments offer improved survival, reduced toxicity, and enhanced quality of life for patients worldwide.

## Figures and Tables

**Figure 1 cancers-17-01980-f001:**
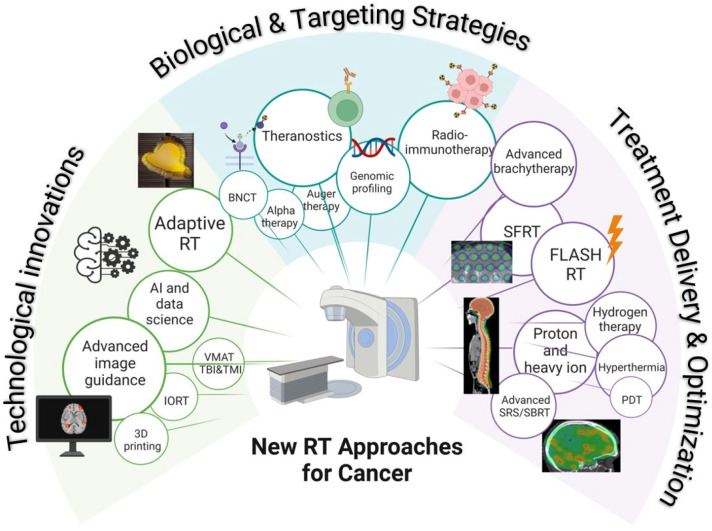
New RT approaches for cancer. Abbreviations in the figure: artificial intelligence (AI), boron neutron capture therapy (BNCT), intraoperative RT (IORT), photodynamic therapy (PDT), spatially fractionated RT (SFRT), three-dimensional (3D), total body irradiation (TBI), total marrow irradiation (TMI), volumetric modulated arc therapy (VMAT).

**Figure 2 cancers-17-01980-f002:**
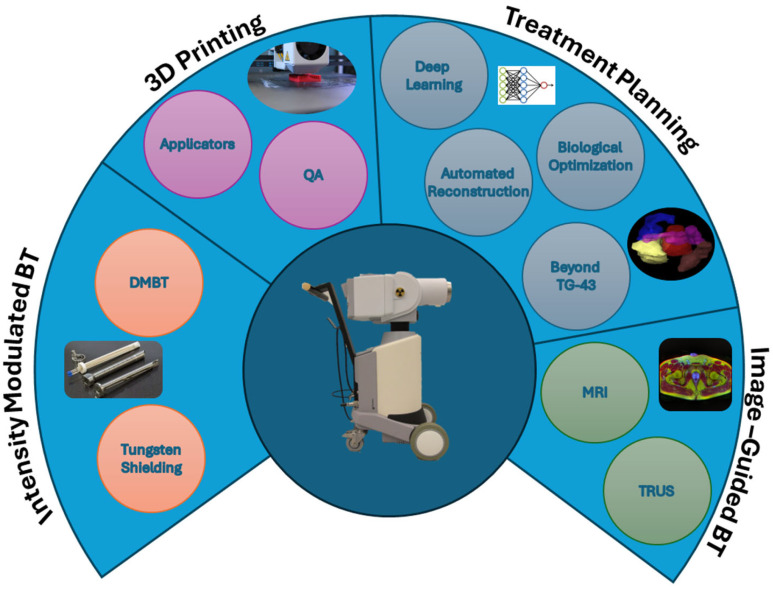
New approaches in brachytherapy. Abbreviations in the figure: dynamic modulated brachytherapy (DMBT), quality assurance (QA), magnetic resonance imaging (MRI), transrectal ultrasound (TRUS).

**Figure 3 cancers-17-01980-f003:**
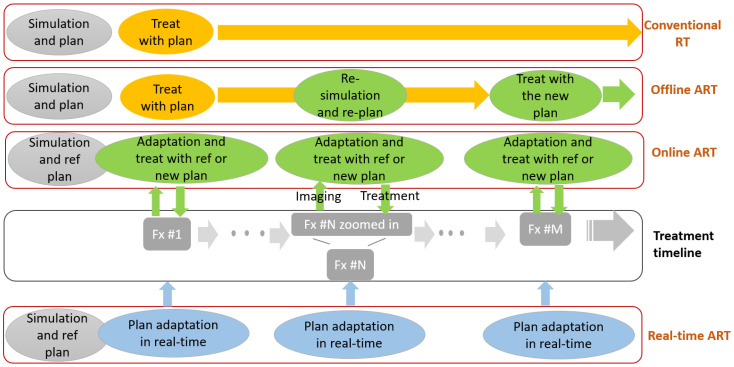
A schematic comparing conventional RT, offline ART, online ART, and real-time ART. For online and real-time ART, the initial simulation is used to develop a reference (ref) plan.

**Figure 4 cancers-17-01980-f004:**
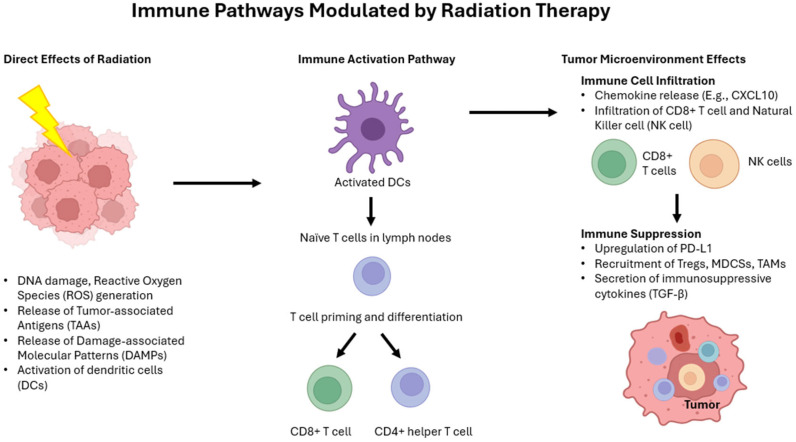
Immune modulation by radiation therapy. Radiation induces both immunostimulatory and immunosuppressive effects through pathways involving dendritic cell activation, T cell priming, chemokine-driven immune cell infiltration, and the recruitment of immunosuppressive cells within the tumor microenvironment.

**Figure 5 cancers-17-01980-f005:**
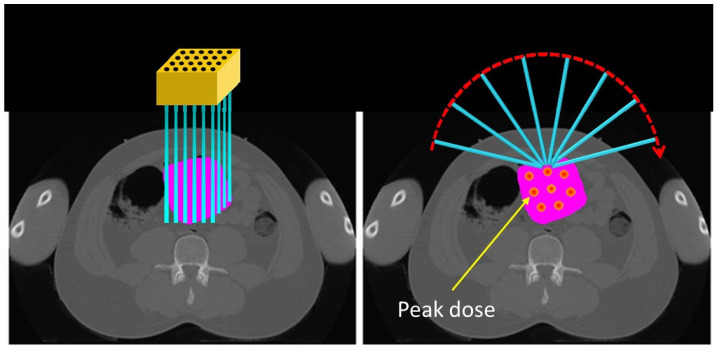
Conceptual comparison between the traditional 2D GRID therapy (**left**) and the 3D LATTICE therapy (**right**).

**Figure 6 cancers-17-01980-f006:**
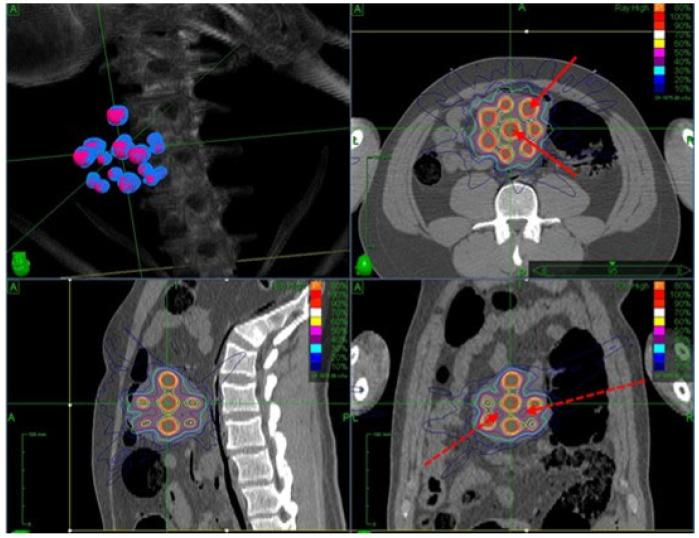
A simulated SFRT plan with peak dose 15 Gy per fraction. Solid and dash arrows indicate the peak and valley dose regions, respectively. (Multiplan TPS 5.2.1 for CyberKnife).

**Table 1 cancers-17-01980-t001:** List of FDA approved theranostics.

Agents (Trade Name)	Therapy (Imaging)	Trial	Dosage	Indications
PSMA [210] (Locametz)	^177^Lu (^68^Ga)	VISION	7.4 GBq every 6 weeks up to 6 doses.	PSMA metastatic castration-resistant prostate cancer
MIBG [211] (Azedra)	^131^I (^123^I)	IB12B	18.5 GBq or 296 MBq/kg, based on weight	Pheochromocytoma or paraganglioma
DOTATATE [212] (Luthathera)	^177^Lu (^177^Lu)	NETTER-1	7.4 GBq over 30 min every 8 weeks (4 doses total)	Somatostatin receptor-positive gastroenteropancreatic neuroendocrine tumor
NaI	^131^I (^123^I)		3.7–5.55 GBq	Malignant and benign thyroid cancer

**Table 2 cancers-17-01980-t002:** Agents under clinical investigation (as of July 2024).

Agent Therapy (Imaging)	Institute	Study Objective	Trial Identifiers	Target	Indication
^177^Lu (^68^Ga) ^177^Lu (^68^Ga) SABR + ^177^Lu (^68^Ga) ^177^Lu (^68^Ga) ^177^Lu-ITG-PSMA-1 (^68^Ga-PSMA-11) ^68^Ga-DOTA-5G (^177^Lu-DOTA-ABM-5G) ^177^Lu-TLX591 (^68^Ga-PSMA-11) ^212^Pb-VMT01 (^203^Pb-VMT01 or ^68^Ga-VMT02) ^212^Pb-VMT-α-NET (^203^Pb-VMT-α-NET)	Centre Leon Berard, France CIUSS de l’Estrie-CHUS hospital, Canada Peter MacCallum Cancer Centre, Australia St. Olavs hospital, Norway University of Laussanne Hospitals (Switzerland) University of California, Davis, USA Telix Pharmaceuticals Pty Limited Perspective therapeutics University of Iowa	Safety and efficacy Eligibility for endoradiotherapy SABR alone vs. SABR + ^177^Lu-PSMA Improve diagnostic and therapeutic glioma management Feasibility study Safety and efficacy Safety, Biodistribution, and dosimetry Safety and efficacy Safety and efficacy	NCT06059014 (Phase I/II) NCT05214820 (Phase II) NCT05560659 POPSTAR II (Phase II) NCT05644080 NCT05420727 NCT06389123 NCT04786847 (ProstACTSelect) NCT05655312 NCT06148636	PSMA PSMA PSMA PSMA Vasculature of soft tissue sarcoma αvβ6integrin PSMA Melanocortin subtype 1 receptor Somatostatin receptor type 2	Metastatic clear cell renal cancer Upper Gastrointestinal cancer Oligometastatic prostate cancer Recurrent Grade 3 and 4 Glioma Soft tissue sarcoma Metastatic carcinomas, Pancreatic Cancer, non-small lung cancer Metastatic castration-resistant prostate cancer Unresectable and metastatic melanoma Neuroendocrine tumors

**Table 3 cancers-17-01980-t003:** FDA approved radioimmunotherapy with its respective cancer treatment and RT indications.

Types	Immune Drugs	RT Indications	Cancers
PD-1/PD-L1 inhibitors	Pembrolizumab, Nivolumab, Atezolizumab, Avelumab, Durvalumab	Boost immune responses	NSCLC, GBM, bladder
CTLA-4 inhibitors	Ipilimumab	Enhance systemic anti-tumor responses	Melanoma, NSCLC
TGF-β inhibitors	Fresolimumab, LY2109761, LY364947	Modulate the tumor microenvironment and enhance immune responses/radio-resistance	NSCLC, metastatic breast, GBM
Cytokines	GM-CSF	Abscopal effect	Laryngeal

**Table 4 cancers-17-01980-t004:** Common dosimetric parameters used in different SFRT techniques.

	Peak Dose	Beam Size	Beam Space	Peak Valley Dose Ratio
GRID Therapy	10–20 Gy	1–2 cm	1–4 cm	3–7
LATTICE Therapy	8–20 Gy	1–2 cm	1–4 cm	3–7
MBRT	50–100 Gy	0.3–1 mm	1–4 mm	10–20
MRT	300–600 Gy	25–100 µm	200–400 µm	>50

**Table 5 cancers-17-01980-t005:** Comparison of SFRT techniques: GRID, LATTICE, MBRT, and MRT. The table summarizes key differences and overlaps among the four approaches.

	GRID	LATTICE	MBRT	MRT
Clinical Readiness	Limited	Limited and early phased	Preclinical	Preclinical
Beam scale	cm scale 2D	cm scale 3D	mm scale	μm scale
Dose delivery	Static	VMAT/IMRT	Preclinical systems	Preclinical systems
Need Grid/Slit Collimator?	Yes	No	Yes	Yes
Need SFRT structure contour?	No	Yes	No	No

**Table 6 cancers-17-01980-t006:** Select clinical studies evaluating GRID and LATTICE therapy.

	Clinical Trials				
	Authors	Number of Patients	Primary Tumor Site	SFRT Rx	Control Rate
GRID therapy	Mohiuddin et al., 1996 [272]	87	Diverse (including SCC, sarcoma)	10–25 Gy single fx SFRT only, 10–15 Gy single fx SFRT + EBRT	Response rate: 91%
	Huhn et al., 2006 [273]	27	SCC of H&N	15–20 Gy single fx SFRT + EBRT	Neck control: 92–93%
LATTICE therapy	Larrea et al., 2022 [279]	11	NSCLC	15 Gy single fx SFRT + EBRT	Complete response 18.2%, response > 50% is 54.4%
	Amendola et al., 2020 [281]	10	Cervix Cancer	24 Gy in three fx SFRT + EBRT	Complete metabolic response: 88.9%

**Table 7 cancers-17-01980-t007:** Comparative analysis of established and emerging radiation therapy techniques covered in detail. This table provides a summary of the major advantages, disadvantages, clinical readiness, and key challenges associated with core radiation therapy approaches discussed in detail in the manuscript. Techniques included represent both current clinical standards and cutting-edge innovations within therapeutic radiology.

Technique	Advantages	Disadvantages	Clinical Readiness	Key Challenges
**Stereotactic RT**	High precision, few fractions, established efficacy	Requires advanced imaging, not suitable for all tumors	Routine	Motion management, expanding to new indications
**Brachytherapy**	High local dose, short treatment time	Invasive, declining usage in some regions, procedural, resource-intensive	Routine/ evolving	Training, accessibility
**Advanced Image Guidance**	Enhances precision, real-time targeting, biological and functional imaging	Cost, increased treatment time	Routine	Workflow integration
**Proton/Heavy Ion RT**	Superior dosimetry	High cost, limited availability	Niche	Cost-effectiveness, clinical evidence, still evolving
**Adaptive RT**	Personalized treatment, daily replanning	Resource-intensive, complex workflow	Expanding	Automation, clinician training
**AI/Data Science**	Faster planning, predictive analytics	Data quality, validation, regulatory hurdles	Early Adoption	Transparency, data privacy, generalizability
**Hyperthermia**	Radiosensitization, immune stimulation	Limited availability, specialized equipment, resource-intensive	Niche	Infrastructure, standardization
**Theranostics**	Dual imaging + therapy, systemic/local combo	Radiopharmaceutical logistics	Expanding	Integration with RT workflows
**Radioimmunotherapy**	Systemic effect, synergy with RT	Targeting specificity, toxicity, limited approvals	Niche	Drug delivery, trial validation
**Spatially Fractionated RT**	Possible immune boost, bulk tumor targeting	Planning complexity, experimental	Early Adoption	Lack of guidelines, validation
**FLASH RT**	Reduced toxicity, ultrafast delivery	Technological challenges	Experimental	Dose control, machine availability
**Boron Neutron Capture Therapy**	Tumor-selective targeting	Requires neutron sources, complex dosimetry	Experimental	Infrastructure, boron delivery agents

**Table 8 cancers-17-01980-t008:** Summary of adjunct and experimental modalities briefly covered in the manuscript. This table outlines the benefits, limitations, clinical readiness, and implementation challenges of additional radiation-related or biologically driven modalities that are mentioned more briefly. These approaches represent emerging or niche technologies relevant to the future landscape of radiation oncology.

Technique	Advantages	Disadvantages	Clinical Readiness	Key Challenges
**VMAT for TBI/TMI**	Conformal dose to target, organ sparing	Complex planning, time-intensive	Expanding	Automation, access in resource-limited settings
**Intraoperative RT (IORT)**	One-time RT during surgery, reduces recurrence	Specialized equipment, limited tumor sites, procedural	Niche	Logistics, standardization
**Genomic Profiling**	Personalized therapy, predicts RT response	Interpretation complexity, limited RT-specific use	Early Adoption	Validation, integration with RT decisions
**3D Printing**	Personalized bolus/applicators, QA tools	Material cost, regulatory approval for devices	Expanding	Standardization, clinical outcome validation
**Photodynamic Therapy**	Non-ionizing, minimal systemic toxicity	Depth limitation, photosensitivity	Niche	Agent development, tumor accessibility
**Alpha-Particle Therapy**	High LET, short range, ideal for micrometastases	Limited isotopes, toxicity control	Expanding	Delivery, safety, cost
**Auger Therapy**	High LET at DNA-level, minimal bystander damage	Inefficient targeting, low efficacy so far	Preclinical	Compound design, delivery to DNA
**Hydrogen Therapy**	Radioprotective, antioxidant potential	Unproven in clinical oncology	Preclinical	Mechanistic validation, delivery systems

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
