# Peer review of "New Approaches in Radiotherapy"

_cancers, 2025, doi:10.3390/cancers17121980_

Round 1
Reviewer 1 Report
Comments and Suggestions for Authors
This manuscript is a comprehensive review of recent innovations in radiotherapy (RT), emphasizing how technological, biological, and computational advancements are shaping the future of cancer treatment. It covers adaptive RT, advanced image guidance systems (including MRI and PET-based platforms), and the integration of artificial intelligence in imaging, treatment planning, and quality assurance. The paper also explores emerging therapeutic techniques such as FLASH RT, spatially fractionated RT (SFRT), boron neutron capture therapy (BNCT), and radioimmunotherapy, discussing their biological mechanisms, clinical potential, and ongoing trials. Additionally, it highlights the growing role of personalized medicine through approaches like theranostics and genomics. Overall, the review outlines a dynamic evolution in RT aiming to improve precision, minimize side effects, and enhance therapeutic outcomes. However, several sections of the manuscript require revision and clarification before it can be considered for publication.
- Adaptive Radiotherapy (Section 2): Include more on the clinical impact of ART from large trials.
- Advanced Image Guidance (Section 3):In MR-LINAC and PET-LINAC subsections, provide specific comparative performance data or key studies to strengthen the discussion.
- Artificial Intelligence (Section 4):Add more discussion on validation and regulation challenges, particularly in clinical AI deployment.
- Boron Neutron Capture Therapy (Section 5):Expand on recent clinical trials in Japan or Europe to support the clinical relevance of BNCT.
- Brachytherapy (Section 6):Expand discussion on the clinical outcomes and toxicity profiles for IMBT and 3D-printed applicators.
- FLASH RT (Section 7):Clarify ongoing debates about oxygen depletion vs. immune activation mechanisms.
- Radioimmunotherapy (Section 9):Suggest including one diagram summarizing immune pathways modulated by RT.
- Spatially Fractionated RT (Section 10):Include a critical view on its current limitations and lack of large-scale prospective clinical trials. Also, Clarify differences and overlaps between GRID, LATTICE, MBRT, and MRT with a comparative figure.
- Add a summary figure/table of all discussed techniques. A comparative table showing advantages, disadvantages, clinical readiness, and key challenges for each RT approach (e.g., FLASH, BNCT, AI) would improve clarity.
- Avoid overuse of ‘Error! Reference source not found.
- Update references to include more recent trials and systematic reviews from 2022–2025.
Author Response
Adaptive Radiotherapy (Section 2): Include more on the clinical impact of ART from large trials.
Response 1:
We thank the reviewer for this important suggestion. We have added a discussion at the end of the ART section summarizing findings from recent clinical trials assessing the clinical impact of ART, including outcomes on local control and toxicity reduction.
Comment 2:
Advanced Image Guidance (Section 3): In MR-LINAC and PET-LINAC subsections, provide specific comparative performance data or key studies to strengthen the discussion.
Response 2:
We have incorporated several comparisons between MR-LINAC and PET-LINAC. These additions highlight some of the key differences between them.
Comment 3:
Artificial Intelligence (Section 4): Add more discussion on validation and regulation challenges, particularly in clinical AI deployment.
Response 3:
A dedicated paragraph has been added to the "Current Challenges and Future Outlook" subsection discussing validation and regulatory issues in AI implementation. This includes guidance and frameworks from the WHO, FDA, and IAEA.
Comment 4:
Boron Neutron Capture Therapy (Section 5): Expand on recent clinical trials in Japan or Europe to support the clinical relevance of BNCT.
Response 4:
We have expanded the “Clinical Applications and Future Directions” section of BNCT to include a summary of recent trials from Japan and Europe, focusing on tumor types, treatment outcomes, and trial phases.
Comment 5:
Brachytherapy (Section 6): Expand discussion on the clinical outcomes and toxicity profiles for IMBT and 3D-printed applicators.
Response 5:
Due to the limited availability of published clinical outcome data on IMBT, we have included a new paragraph discussing the likely reasons, including the early stage of clinical implementation and technological constraints. For 3D-printed applicators, we now summarize key findings from existing studies, although we note that data remains limited at present.
Comment 6:
FLASH RT (Section 7): Clarify ongoing debates about oxygen depletion vs. immune activation mechanisms.
Response 6:
We have revised the FLASH RT section to include a discussion of current hypotheses regarding the underlying radiobiological mechanisms, including recent studies examining both oxygen depletion and immune activation. Multiple perspectives are presented to reflect the ongoing debate in the field.
Comment 7:
Radioimmunotherapy (Section 9): Suggest including one diagram summarizing immune pathways modulated by RT.
Response 7:
We have added the recommended diagram to the section. The figure summarizes key immune pathways modulated by radiotherapy, including antigen presentation, immune checkpoint regulation, and tumor microenvironment changes.
Comment 8:
Spatially Fractionated RT (Section 10): Include a critical view on its current limitations and lack of large-scale prospective clinical trials. Also, clarify differences and overlaps between GRID, LATTICE, MBRT, and MRT with a comparative figure.
Response 8:
We now address the limitations of SFRT, including the lack of large-scale prospective data and technical standardization, in the “Current Challenges and Future Directions” section. Additionally, we have included a comparative table distinguishing GRID, LATTICE, MBRT, and MRT based on delivery technique, spatial modulation, and current clinical status.
Comment 9:
Add a summary figure/table of all discussed techniques. A comparative table showing advantages, disadvantages, clinical readiness, and key challenges for each RT approach (e.g., FLASH, BNCT, AI) would improve clarity.
Response 9:
In response to this suggestion, we have added two summary tables. The first focuses on the main areas discussed in the manuscript, while the second outlines additional emerging techniques. Each table includes comparative data on advantages, disadvantages, clinical readiness, and key challenges.
Comment 10:
Avoid overuse of ‘Error! Reference source not found.’
Response 10:
We sincerely apologize for this oversight. The referencing issue occurred during the PDF conversion process and has now been corrected throughout the manuscript.
Comment 11:
Update references to include more recent trials and systematic reviews from 2022–2025.
Response 11:
We appreciate this valuable feedback. Several outdated references have been replaced or supplemented with recent trials and systematic reviews published between 2022 and 2025 to ensure the manuscript reflects the latest developments in the field.
Reviewer 2 Report
Comments and Suggestions for Authors
This is and interesting review paper covering a broad range of radiation treatment modalities. Some of them are real innovative technical approaches while other are tools that can be implemented (AI and data science). Moreover some of them are already generally accepted treatments (SRS/SBRT and IORT) while others are quite new and still experimental techniques (flash and micro/minibeams). I would suggest the authors to organize better the text and decide whether to include only real innovative techniques instead of describing all the technical approaches.
Author Response
Comment 1:
This is and interesting review paper covering a broad range of radiation treatment modalities. Some of them are real innovative technical approaches while other are tools that can be implemented (AI and data science). Moreover some of them are already generally accepted treatments (SRS/SBRT and IORT) while others are quite new and still experimental techniques (flash and micro/minibeams). I would suggest the authors to organize better the text and decide whether to include only real innovative techniques instead of describing all the technical approaches.
Response 1:
We thank the reviewer for their insightful comment and their recognition of the paper's broad scope. We agree that the field of radiation oncology includes a wide spectrum of technologies—ranging from well-established modalities like SRS/SBRT and IORT, to emerging tools such as AI/data science, and still-experimental approaches like FLASH and micro/minibeam therapy.
Our intention with this review was to address a gap we observed during our initial literature search: most existing comprehensive reviews are outdated and do not reflect the current pace of innovation, with some still citing 3D conformal RT and IMRT as recent advances. This motivated us to create a forward-looking, inclusive overview of current innovations—spanning established, evolving, and experimental techniques—authored by a multidisciplinary team.
Rather than categorizing techniques by adoption stage, we chose to highlight innovations across the continuum. For mature techniques, we focused on recent clinical or technical advances. For newer modalities, we emphasized early translational data and emerging evidence. We felt this format would best serve the needs of a diverse readership and reflect how clinical practice, research, and technology often evolve in parallel.
We understand the reviewer’s concern about coherence and organization, and appreciate the suggestion to potentially focus on only the most novel approaches. However, we believe the comprehensive approach better contextualizes innovation within the broader clinical ecosystem. We have reorganized the manuscript to flow from the most established techniques to teh most experimental ones, while acknowledging that overlap exists. Still, if the Editor and Reviewer feel the manuscript would be more impactful if split, we are open to restructuring the content into 2–3 narrower, companion articles with distinct thematic focuses.
Reviewer 3 Report
Comments and Suggestions for Authors
This is an extensive detailed review on new developments in radiation-oncology, very well written and presented.
I have only minor remarks:
In my opinion hyperthermia maybe discussed more prominently as there are currently machines with superficial and deep hyperthermia, with software being able to calculate thermal isodose curves and integrate this into the radiation planning. Hyperthermia is at the edge of being implemented into standard. Maybe this could be moved a bit more forward.
Another issue eventually to be added into future perspectives is the issue of the currently new ICRU version: that geometrical thinking in margins for PTV definition shall be dropped and probabilities of recurrences etc shall be used. This is a sort of revolution.
Details:
Page 2, line 68/69: Typo: "in.."
Page 7, line 255: I personally do not agree, accuracy for cranial targets CANNOT be reached with surface guidance only. It always needs additional imaging for anatomical definition.
Page 11, line 458: Typo "stagesthe success"
Page 11, line 464: Glioblastoma are no longer called (since the last WHO classification) "multiforme". Drop "multiforme".
Author Response
Comment 1:
In my opinion hyperthermia may be discussed more prominently as there are currently machines with superficial and deep hyperthermia, with software being able to calculate thermal isodose curves and integrate this into the radiation planning. Hyperthermia is at the edge of being implemented into standard. Maybe this could be moved a bit more forward.
Response 1:
We appreciate the reviewer’s thoughtful suggestion regarding hyperthermia. In response, we have elevated hyperthermia from the "Other Areas of Interest" section into the main body of the manuscript. This section has been significantly expanded to reflect recent clinical and technological developments.
Comment 2:
Another issue eventually to be added into future perspectives is the issue of the currently new ICRU version: that geometrical thinking in margins for PTV definition shall be dropped and probabilities of recurrences etc shall be used. This is a sort of revolution.
Response 2:
Thank you for highlighting this important development. We have added a brief discussion of the topic in the “Other Areas of Interest” section. While still in early stages of dissemination, we agree this paradigm shift from geometric to probabilistic definitions represents a potentially transformative evolution in treatment planning philosophy.
Comment 3:
Page 2, line 68/69: Typo: "in.."
Response 3:
This was a formatting issue that has been corrected.
Comment 4:
Page 7, line 255: I personally do not agree, accuracy for cranial targets CANNOT be reached with surface guidance only. It always needs additional imaging for anatomical definition.
Response 4:
Thank you for this clarification. We acknowledge that the role of SGRT for cranial targets remains debated. We have revised the sentence to state that SGRT has been used in cranial cases, with appropriate references, while avoiding any suggestion that it is sufficient on its own without supplementary imaging.
Comment 5:
Page 11, line 458: Typo "stagesthe success"
Response 5:
Corrected—thank you for catching this.
Comment 6:
Page 11, line 464: Glioblastoma are no longer called (since the last WHO classification) "multiforme". Drop "multiforme".
Response 6:
We appreciate this correction and have removed “multiforme” in accordance with the latest WHO classification terminology.
Reviewer 4 Report
Comments and Suggestions for Authors
The authors comprehensively reviewed new approaches and advances in radiotherapy. It covered all aspects of new approaches in radiotherapy including technological innovations, biological and targeting strategies, and treatment delivery and optimizations. The current hop spots like AI, BNCT, theragnostic radiotracers in clinical trials are highlighted. It is very informative and well written. I suggest that this paper could be published in the journal Cancers as is.
Author Response
Comments 1: The authors comprehensively reviewed new approaches and advances in radiotherapy. It covered all aspects of new approaches in radiotherapy including technological innovations, biological and targeting strategies, and treatment delivery and optimizations. The current hot spots like AI, BNCT, theragnostic radiotracers in clinical trials are highlighted. It is very informative and well written. I suggest that this paper could be published in the journal Cancers as is.
Response 1:
We sincerely thank the reviewer for their generous and supportive evaluation of our manuscript. We are grateful for the recognition of the breadth and relevance of the topics covered, particularly the inclusion of current high-interest areas such as AI, BNCT, and theragnostic approaches. We also appreciate the kind comments regarding the clarity and quality of the writing. Your recommendation for publication is deeply appreciated by our entire team.
Round 2
Reviewer 1 Report
Comments and Suggestions for Authors
Accept in the current version.